# Damage Location Determination with Data Augmentation of Guided Ultrasonic Wave Features and Explainable Neural Network Approach for Integrated Sensor Systems

Christoph Polle [1,2,*,†,‡], Stefan Bosse [1,‡] and Axel S. Herrmann [1,2]

1   Department of Computer Science, University of Bremen, 28359 Bremen, Germany;
    sbosse@uni-bremen.de (S.B.); herrmann@faserinstitut.de (A.S.H.)
2   Faserinstitut Bremen, Am Biologischen Garten 2—Geb. IW3, D-28359 Bremen, Germany
*   Correspondence: cpolle@uni-bremen.de; Tel.: +49-(0)421-218-59662
†   Current address is the affiliation 2.
‡   These authors contributed equally to this work.

**Abstract:** Machine learning techniques such as deep learning have already been successfully applied in Structural Health Monitoring (SHM) for damage localization using Ultrasonic Guided Waves (UGW) at various temperatures. However, a common issue arises due to the time-consuming nature of collecting guided wave measurements at different temperatures, resulting in an insufficient amount of training data. Since SHM systems are predominantly employed in sensitive structures, there is a significant interest in utilizing methods and algorithms that are transparent and comprehensible. In this study, a method is presented to augment feature data by generating a large number of training features from a relatively limited set of measurements. In addition, robustness to environmental changes, e.g., temperature fluctuations, is improved. This is achieved by utilizing a known temperature compensation method called temperature scaling to determine the function of signal features as a function of temperature. These functions can then be used for data generation. To gain a better understanding of how the damage localization predictions are made, a known explainable neural network (XANN) architecture is employed and trained with the generated data. The trained XANN model was then used to examine and validate the artificially generated signal features and to improve the augmentation process. The presented method demonstrates a significant increase in the number of training data points. Furthermore, the use of the XANN architecture as a predictor model enables a deeper interpretation of the prediction methods employed by the network.

**Keywords:** neural networks; explainable AI; feature extraction; data augmentation; Ultrasonic Guided Waves; Structural Health Monitoring

## 1. Introduction and Motivation

Structural Health Monitoring (SHM) is a method to derive features $y$ of mechanical structures and components from measuring signals $x$ (the input features), i.e., SHM is a function $f(x) : x \rightarrow y$. The features as the output of an SHM system can be divided into different levels and classes, giving information about the structural state [1]:

1.  Binary damage indicator feature (there is a damage or defect, negative training);
2.  Position feature (localization, in addition to classification) (where something is);
3.  Multi-class damage or defect feature (we can distinguish different damage classes);
4.  Maintenance level feature (when to do; priority and time prediction);
5.  Lifetime and fatigue prediction features (how long we can use it; time prediction);

In this work, we address binary damage and position feature levels. There is a broad range of different measuring technologies that can be used as the input $x$ of the damage diagnostics SHM system, for example, strain measured by strain gauges [2,3], or acoustic

emission (AE) data received from piezo-electric materials or by fiber optic acoustic emission sensors [4,5]. For plate-like structures, which are the target of this investigation, Ultrasonic Guided Waves (UGW) are of great importance [6] and have garnered significant attention in both research and industry for SHM systems. In plates, UGW occurs as Lamb waves, which were mathematically first described by Horace Lamb [7]. The UGW can be used in AE techniques or, as in the present work, in a pitch catch measurement system using piezo-electric transducers for signal actuation and sensing. UGW offers the advantage of investigating a large area for damage using a limited number of sensors [8,9].

A major problem for SHM systems that use UGW are environmental influences that alter the UGW signal and thus could lead to false damage detection. Due to this, the amount of training data necessary to accurately localize damage with UGW is huge. The environmental factor with the most notable effect on UGW signals is temperature [10,11]. In the past, many researchers have addressed this problem and developed temperature compensation methods. At first, Lu and Michaels [12] proposed the Optimal Baseline Selection (OBS) method, in which a large library of baseline signals at different temperatures is stored. A current measured signal is then compared to the best-fitting baseline signal. The big drawback of OBS is the large amount of data needed to store all the baseline signals, which makes the practical application of this method difficult. To address this problem, the Baseline Signal Stretch (BSS) method was developed and refined in multiple studies [13–15]. In BSS, the UGW baseline signal is stretched to compensate for the influence of temperature on the signal before it is compared to a currently measured signal. Since the effect of UGW signal stretching and contracting increases with the increasing temperature and propagation length (and time) [12,16], BSS is most suitable for small sensor distances and small temperature changes. This and the high computational cost of the signal stretching process make it hard to apply BSS in real-world scenarios.

Douglas and Harley [17] developed another method that is based on Dynamic Time Warping (DTW). Here, the UGW signals were stretched and compressed locally to align the baseline signal as close as possible to a currently measured signal. This method also has some drawbacks, since, due to overfitting damage, information in the signal may get lost. Also, this method is computationally expensive.

Data-driven methods have also been developed for temperature compensation, as demonstrated by Fendzi et al. in [18], where a temperature-dependent phase and amplitude factor for the first UGW wave package were determined from the signal Hilbert envelope. These factors were then used to replicate the wave package at a certain temperature by using the Hilbert envelope of a baseline signal and correcting the phase and the amplitude with the estimated factors.

Azuara and Barrera used a similar approach in [19]. They also extracted at specific temperatures from the signal envelope temperature-dependent phase and amplitude factors using polynomial regression. These factors were then used to build new baseline signals at the current temperature by adjusting phase and amplitude. Nan and Aliabadi, on the other hand, used temperature-dependent Time of Flight (ToF) and amplitude factors in [20] to construct new baseline signals. In this method, they used amplitudes and ToF extracted from the maxima and minima of the first wave package of signals measured at different temperatures to estimate the compensation factors with cubic regression. The construction of new baseline signals was achieved via a multiplication of the amplitude factor with the initial baseline, by a multiplication of the ToF factor with the time array of the initial baseline array, and finally, by a linear interpolation of the initial baseline signal. Till now, all the data-driven methods mentioned have in common that they use temperature-dependent compensation factors to generate new baseline signals. Since these methods are computationally expensive and require the storage of at least some baseline measurements to create new baseline signals, they are impractical for scenarios where the hardware has limited resources, as in sensor nodes that were integrated into the material under investigation.

Another data-driven method, called Temperature Scaling (TS), was introduced in [21] to address these issues. TS uses particular features in a UGW signal, like the maximum $M$ of the UGW signal envelope, which can be expressed as functions $M(T)$ of the temperature $T$. These feature functions are used for temperature compensation and damage detection and leads to a reduction in all UGW signal points to a single value. This huge reduction and simplification of UGW data enables the possibility of using them in sensor nodes integrated into the material. The features used in TS have also been successfully used for damage localization by using machine learning algorithms like artificial neural networks (ANN) [22]. But, UGW measures are time-intensive, especially when measured under varying temperatures, which often leads to an insufficient number of data for effective model training and validation, as can be seen in [22], for example. Due to the problem of measuring a sufficient number of UGW data, simulated data is often used for training ANNs of all kinds.

In the course of the investigations of TS, it was seen that the UGW signal envelope maximum could also be expressed as a function $M(T)$, even if damage was present. As a result of this, the idea arose to use the feature functions to generate synthetic feature data for machine learning model training like ANN. These models could then be validated with a test dataset containing only original feature data.

There is a great interest in developing SHM systems that have comprehensive and interpretable data analysis since SHM is primarily used for critical structures. On the other hand, ANN models are often referred to as "black boxes" since their prediction function is unknown from an analytical or numerical point of view, which makes it difficult to understand the underlying decision-making processes in an ANN. An ANN is, in general, nothing else but a parameterizable functional graph that consists of linear and non-linear function nodes. The composition of the parameter can result in highly non-linear functions that cannot be accurately represented by a polynomial. An ANN has a transfer function $f(\vec{x}, \vec{p}) : \vec{x} \rightarrow \vec{y}$ which maps a multi-dimensional input vector $\vec{x}$ to a scalar or vector output $\vec{y}$. Due to random parameter initialization ($\vec{p} \leftarrow rand$), every training run can result in differing transfer functions, which complicates any attempt at interpretation. Additionally, an ANN can be divided into layers, where generally the outputs of all nodes $n_{i,j}$ in one layer $i$ are connected to each node of the following layer $j = i + 1$. This full connection eliminates very useful internal interpretations of the ANN model.

To address these issues, Lomazzi et al applied in [23] the Layer-wise Relevance Propagation (LRP) algorithm [24] to a CNN to create an Explainable Artificial Intelligence (XAI) for damage detection using UGW. The method identifies the most relevant features in each layer and, with that, the most important transducer paths. The LRP algorithm does that by calculating a relevance value for all nodes in a layer of the CNN by using a backpropagation method. The relevance values show the contribution of all nodes in each layer to the inputs of the output layer. In contrast to conventional gradient-based backpropagation, which is used for model training, LRP is designed to relate the predictions of an ANN to specific input features. With this information, it is possible to determine which inputs essentially influence the prediction of the neural network. The physical interpretation of LRP results made it possible to understand why CNN used a certain feature. On the other hand, the processing of the features within the CNN and the CNNS transfer function could not be achieved. Thus, the decision-making process of the CNN could not be investigated, which limits the interpretability and explainability of the CNN models.

In [25], the Automated ML Toolbox for Cyclic Sensor Data and the ML Toolbox DAV$^3$E were used for feature extraction and feature selection for Machine Learning (ML) approaches by Schnur et al. Regarding feature selection, the use of these toolboxes enabled a rather limited degree of interpretability but did not provide an insight into the decision-making processes of the models; thus, no interpretability or explanation of the ML models was possible.

In this study, we apply an explainable neural network (XANN) approach to explore the input and output feature dependencies in the UGW data, as outlined in Vaughan et al. [26].

In the XANN architecture used, the network is divided into subnetworks in such a way that it is possible to analyze the inputs and outputs of the individual subnetworks. Due to this, the transfer functions of the subnetworks can be determined, while the transfer function of the overall network remains unknown. This approach enables a step-by-step investigation of the decision-making process within the XANN, which significantly enhances the understanding and interpretability of the decision-making process, especially with regard to UGW feature data. It helps to find the feature data with the most influence on the individual subnetworks. Also, with this method, the validation of artificially generated data will be shown. The work flow of the present study is shown in Figure 1 as an organizational chart.

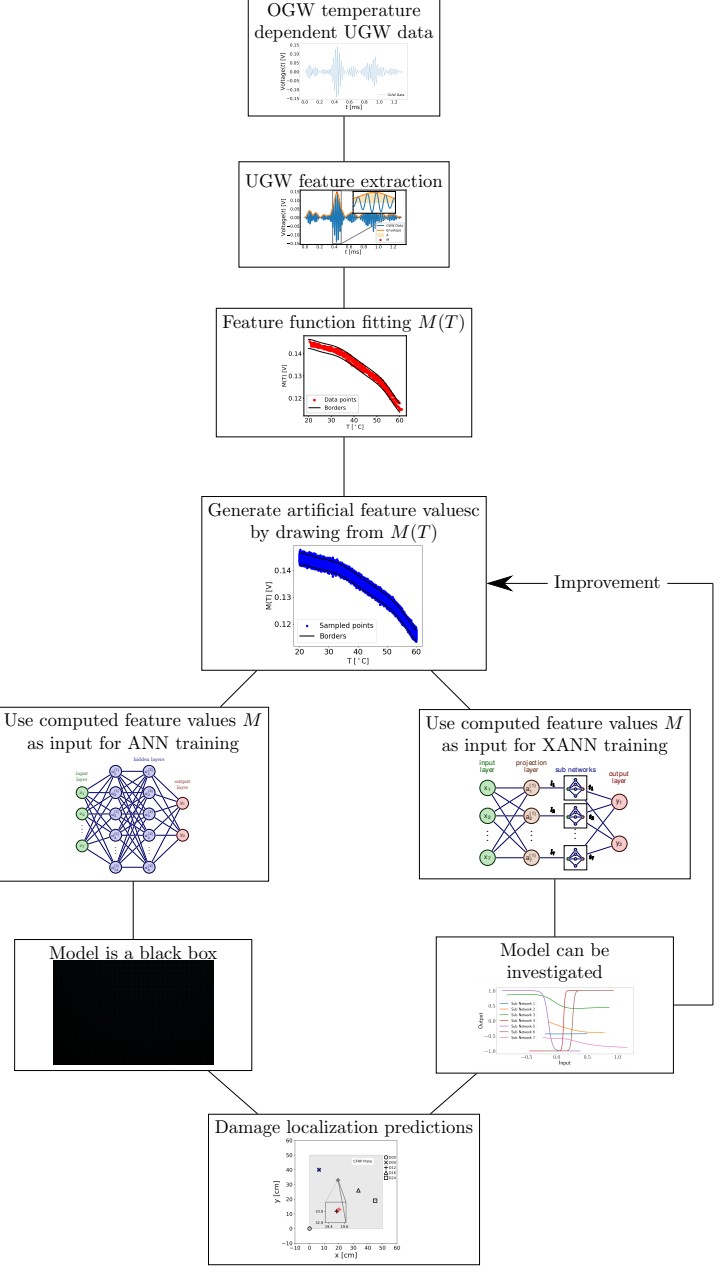

**Figure 1.** Organizational chart of the workflow in the present work.

## 2. Measurement Technologies, Sensor Data, and Feature Extraction

### 2.1. Experimental UGW Datasets

In this study, we use an open dataset [27] provided by the Open Guided Waves (OGW) platform. The dataset consisted of UGW pitch–catch measurements recorded in a temperature range of 20 °C to 60 °C, with temperature steps of 0.5 °C. For each target temperature, pitch–catch measurements were performed using a narrowband 5-cycle Hann-filtered sine wave pitch signal covering center frequencies ranging from 40 kHz to 260 kHz in 20 kHz steps. For the purpose of our study, we specifically focused on signals recorded at a center frequency of 40 kHz. This choice was motivated by the fact that these signals have been found to be the most sensitive to damage [21].

All measurements were conducted on a carbon-fiber-reinforced plastic (CFRP) plate with dimensions of 500 mm × 500 mm × 2 mm. The CFRP plate, made from Hexply ®M21/34%/UD134/T700/300 carbon pre-impregnated fibers, featured a stacking configuration of

$$[45°/0°/-45°/90°/-45°/0°/45°/90°]_S.$$

To facilitate the pitch–catch measurements, 12 DuraAct piezoelectric transducers were co-bonded to the plate, while four artificial damage locations were introduced at different positions on the surface to simulate damage [27]. These simulated damage consisted of circular aluminum disks, 1 cm in diameter, affixed to the CFRP plate using tacky tape. A schematic diagram illustrating the plate setup, including the considered transducer paths with actuator transducers $i \in 1, 2, 3, 4, 5, 6$ and sensor transducers $j \in 7, 8, 9, 10, 11, 12$ with $j = i + 6$, can be found in Figure 2.

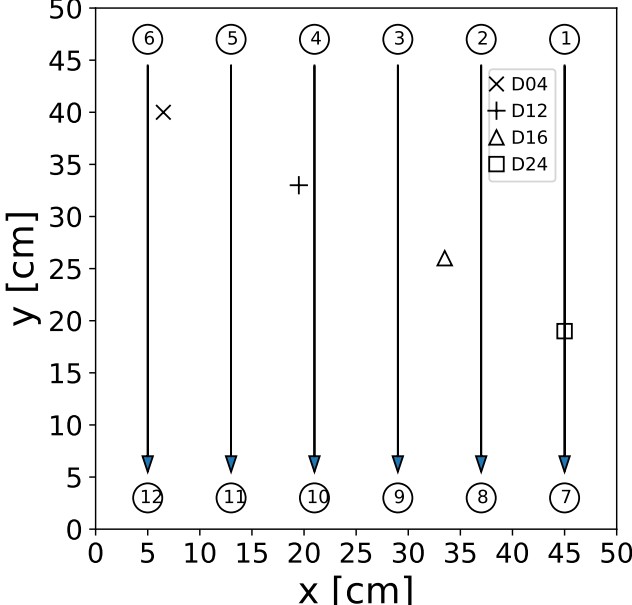

**Figure 2.** Sketch of the CFRP plate used for UGW measurements. The circles with numbers represent the transducer positions; numbers correspond to the transducer indices. The arrows show the transducer paths used in this study. While the D04 (×), D12 (+), D16 (△), and D24 (□) markers correspond to the position of the artificial damage applied to the surface of the plate.

A sketch of the experimental setup is shown in Figure 3. In the picture, it is shown that the plate is placed in a climate chamber. A computer sets the waveform of the pitch signals and sends it to a waveform generator, which passes the signal to an amplifier. The amplifier sends the signal to the actuator. The sensor detects the UGW signal, which is recorded by an oscilloscope. The current temperature of the UGW measurement is recorded by a data logger. The UGW signal and temperature are then passed to the computer where the data is stored.

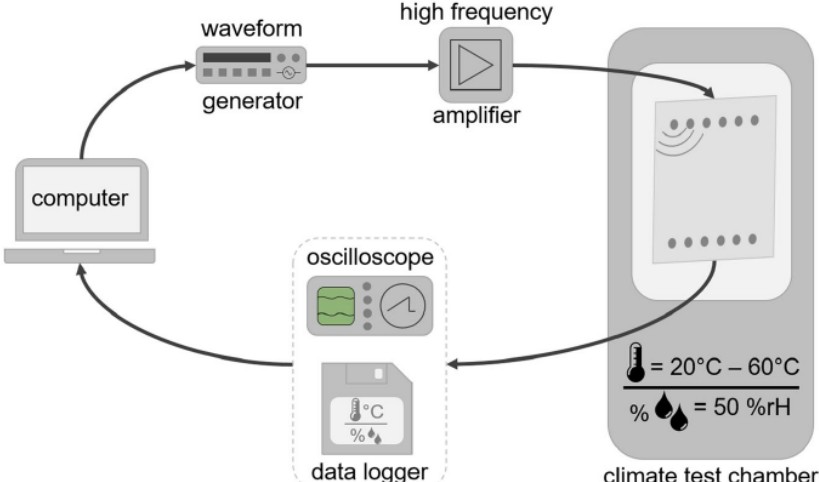

**Figure 3.** A sketch of the experimental setup utilized for the UGW measurements. The image is sourced from [27] under the Creative Commons Attribution 4.0 International License (http://creativecommons.org/licenses/by/4.0/ (accessed on 10 January 2024)).

*2.2. Feature Extraction and Feature Function Determination*

In the study [21], it was demonstrated that certain features extracted from UGW signals, when expressed as functions of temperature, can be effectively used for damage detection. This approach, termed temperature scaling (TS), involves defining reasonable upper and lower borders within which a feature extracted from a currently recorded signal at a specific temperature should fall in order to be considered undamaged. Any feature values that exceed these borders are detected as indications of damage.

Among the various features investigated, the maximum amplitude of the envelope, denoted as $M$, was found to be particularly effective for damage detection. To extract $M$, the analytical signal of the data was first computed. For a given time-dependent signal $s(t)$, the analytical signal $s_\mathrm{a}(t)$ is obtained as

$$s_\mathrm{a}(t) = s(t) + \mathrm{i}s(t) * \frac{1}{\pi t} \tag{1}$$
$$= s(t) + \mathrm{i}\mathcal{H}\{s(t)\} \tag{2}$$

where i represents the imaginary unit, $*$ denotes the convolution operator, and $\mathcal{H}\{\}$ represents the Hilbert transform. The envelope $E_\mathrm{s}(t)$ of the data was then calculated using the analytical signal $s_\mathrm{a}(t)$:

$$E_\mathrm{s}(t) = |s_\mathrm{a}(t)|. \tag{3}$$

Finally, the maximum value of the data envelope, $M$, was determined. Figure 4 illustrates an example of a UGW signal with its corresponding envelope and the extracted maximum.

In Figure 4, the area under the envelope $A$ is also shown since it was used for data normalization.

In the previous study [21], the temperature-dependent functions $M(T)$ were determined solely for the damage-free measurements. However, for the purpose of generating artificially generated data in this work, the functions $M(T)$ were also determined for the damage measurements to enable their use in the analysis.

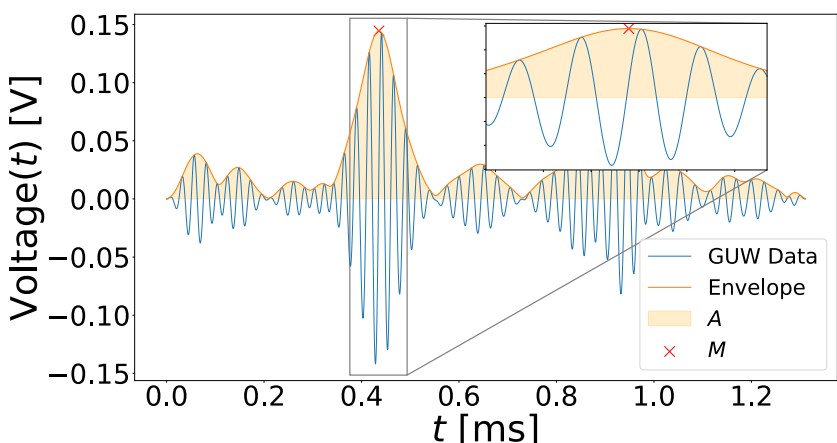

**Figure 4.** Example for maximum $M$ and area under the envelope $A$ extraction from a UGW signal. The displayed signal was recorded at sensor path $1 \rightarrow 7$ with a frequency of 40 kHz at a temperature of 20 °C.

To determine the functions $M(f, T)$ for different transducer paths $i \rightarrow j$, eight-degree polynomials were fitted to the $M$ data:

$$M^{i,j}(T) = \sum_{k=0}^{8} a_k^{i,j} T^k, \tag{4}$$

where $M^{i,j}$ represents the function for transducer path $i \rightarrow j$. The polynomial coefficients $a_k^{i,j}$ were computed to best fit the data.

The temperature-dependent upper boundary $\Delta M_{\text{up}}^{i,j}(T)$ and lower boundary $\Delta M_{\text{low}}^{i,j}(T)$ are defined as

$$\Delta M_{\text{up}}^{i,j}(T) = M^{i,j}(T) + \delta M_{\text{up}}^{i,j}, \tag{5}$$

$$\Delta M_{\text{low}}^{i,j}(T) = M^{i,j}(T) - \delta M_{\text{low}}^{i,j}. \tag{6}$$

where $\delta M_{\text{up}}^{i,j}$ and $\delta M_{\text{low}}^{i,j}$ are constants which are added or subtracted from the functions $M(f, T)$ to define the borders $\Delta M_{\text{up}}^{i,j}(T)$ and $\Delta M_{\text{low}}^{i,j}(T)$.

To calculate $\Delta M_{\text{up}}^{i,j}(T)$ and $\Delta M_{\text{low}}^{i,j}(T)$, the initial values of the constants $\delta M_{\text{up}}^{i,j}$ and $\delta M_{\text{low}}^{i,j}$ were set to zero. Subsequently, each measurement $M^{i,j}$ at its corresponding temperature $T$ was checked to determine if it fell within the range of $\Delta M_{\text{up}}^{i,j}(T)$ and $\Delta M_{\text{low}}^{i,j}(T)$. If a value $M^{i,j}$ at temperature $T$ was found to be smaller than $\Delta M_{\text{low}}^{i,j}(T)$, a very small value of $1 \times 10^{-6}$ was added to $\delta M_{\text{low}}^{i,j}$. Similarly, if a value $M^{i,j}$ at temperature $T$ was larger than $\Delta M_{\text{up}}^{i,j}$, the same small value of $1 \times 10^{-6}$ was added to $\delta M_{\text{up}}^{i,j}$. This iterative process continued until all the data points $M^{i,j}$ within the dataset adhered to the predefined boundaries. Therefore, $\Delta M_{\text{up}}^{i,j}$ and $\Delta M_{\text{low}}^{i,j}$ denote the maximum acceptable deviations of a value $M^{i,j}$ at a particular temperature $T$ from the function $M^{i,j}(T)$, in order to be identified as damage-free. An example of the extracted $M$ values with the determined borders for the damage-free case of transducer path $1 \rightarrow 7$ is shown in Figure 5. It can be observed that all the $M$ data points lie within the upper and lower boundaries.

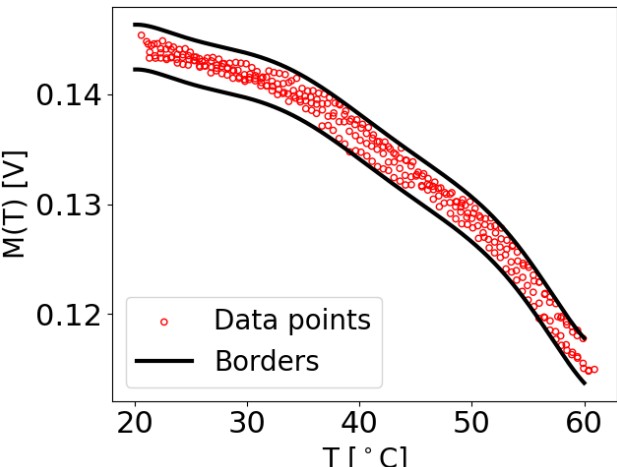

**Figure 5.** Examples of $M$ values extracted from the original data of the OGW dataset, which were used to establish the borders from the maximum function $M^{i,j}(T)$, specifically for the damage-free measurement for the case where $i = 1$ and $j = 7$.

## 3. Data-Driven Modeling

### 3.1. Data-Driven Modeling with ANN and XANN Architectures

The main focus of this work is the model-assisted deployment of data-driven predictor models. In the next subsections, we will discuss the fundamentals of our data models, the missing explainability of black-box models, and a first step towards partial explainability using dedicated model architectures. The data-driven predictor models can be used to provide damage classification and localization as well as support for the synthetic data augmentation discussed in Section 4. We do not address data-driven generative models that create synthetic sensor data. The XANN architecture satisfies multiple goals:

1.  An architecture that allows us to investigate the relationship between input and output variables (only partially possible, but improved compared with ANN);
2.  Acts as a predictor model architecture;
3.  Identification of critical paths inside the network, e.g., proving noise robustness (discussed in Section 3.3);
4.  Improvement and support of the model-assisted synthetic data augmentation.

### 3.1.1. Data Normalization and Scaling

Before being used as training or test data, the envelope maximum $M^{i,j}(T)$ of different transducer paths $i \rightarrow j$ were normalized. The simplest normalization could be achieved by dividing $M^{i,j}(T)$ with the maximum amplitude measured for all temperatures. In the present work, normalization was performed by using the area under the envelope $A^{i,j}$, measured at damage-free. Since the $A^{i,j}(T)$ values just changed slightly across the entire temperature range, an average value $\bar{A}^{i,j}$ was calculated for each transducer path $i \rightarrow j$ and used for the normalization process.

$\bar{A}^{i,j}$ was used for normalization because it captures particular signal characteristics and thus helps in the generalizing process of trained models. This assumption is supported by Figure 6, where one can see that $\bar{A}^{i,j}$ follows a pattern in which the values of $\bar{A}^{i,j}$ are small at the edges of the plate and increases to the middle. This shows that $\bar{A}^{i,j}$ holds important information related to the positions of the signal transducer path.

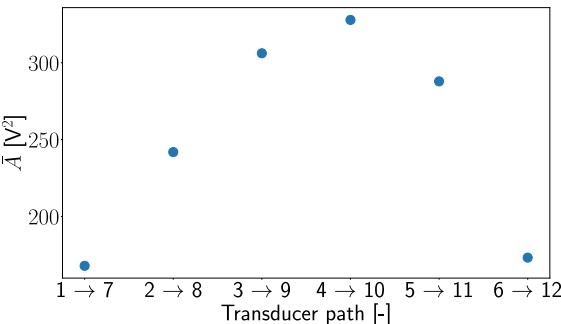

**Figure 6.** Average values of the area under the envelope $\bar{A}^{i,j}$ for different transducer paths for the damage-free case.

Using $\bar{A}^{i,j}$, the normalized maximum values $\hat{M}^{i,j}(T)$, are determined by the following equation:

$$\hat{M}^{i,j}(T) = \frac{M^{i,j}(T)}{\bar{A}^{i,j}} \tag{7}$$

For the neural networks, an input array $I$ in which $T$ and all $\hat{M}^{i,j}(T)$ values of transducer paths $i \rightarrow j$ are stored is used as input:

$$I = [T, \hat{M}^{1,7}(T), \hat{M}^{2,8}(T), \hat{M}^{3,9}(T), \hat{M}^{4,10}(T), \hat{M}^{5,11}(T), \hat{M}^{6,12}(T)]. \tag{8}$$

All elements of $I$ were scaled in such a way that the entries lie in a range of 0 to 1. The scaling rule to achieve this result is as follows:

$$\tilde{I}_k^p = \frac{I_k^p - \min(I_k)}{\max(I_k) - \min(I_k)}, \tag{9}$$

where $k$ represents the element index of $I$, while $p$ represents the index of the current value within that input element. $\tilde{I}_k^p$ corresponds to the scaled value of the current input element, and $\min(I_k)$ and $\max(I_k)$ correspond to the minimum and maximum values of the input element.

Since the models used supervising learning, the artificial damage positions in the x- and y-direction are used as labels $L$, whereas in the case where no damage was present, the labels were defined as x = y = 0. The lanes for the damage case, on the other hand, were scaled to values between 0.2 and 1. The resulting scaling rule states

$$\mathcal{L} = \begin{cases} 0 & \text{if: damage-free case} \\ 0.8 \times \frac{L-65}{450-65} + 0.2 & \text{else} \end{cases} \tag{10}$$

Here, $\mathcal{L}$ donates the scaled x or y label, while $L$ refers to the unscaled x or y label. As can be seen in Figure 2, the value 450 corresponds to the maximal measured damage position and 65 to the minimal measured damage position in the x-y-plane (both measured in mm).

It is important to emphasize that by artificially assigning [0,0] to the undamaged cases, the need for "rescaling" to determine the damage position is eliminated for these cases, so that the direct network outputs are used instead.

### 3.1.2. ANN: The Black Box Model

To showcase the effectiveness of using artificially generated data for training ANNs, a classic ANN architecture was utilized. The architecture consists of an input layer with seven nodes, one node for each computed input feature. This layer is followed by two hidden layers, each with 14 and 8 nodes, respectively, using the tangent hyperbolic function

as the activation function. This function was used because it is non-linear and can be used to separate different classifications in a non-linear, separable space.

The architecture concludes with an output layer comprising two nodes, providing a binary classification and regression of potential damage (or pseudo-defect in our case). Details can be found in [22]. Since the output feature variables (damage position) were scaled between zero and one, the sigmoid activation function was utilized to predict the X- and Y-positions of the damage. The relative positions of a structure under test were scaled and shifted to a range of [0.2,1.0]. All output values below $x < 0.2$ and $y < 0.2$ indicate an undamaged case; all other x-y values above the 0.2 threshold are considered as a damage, providing a combined Classification And Regression (CAR) model. A visual representation of the architecture can be observed in Figure 7.

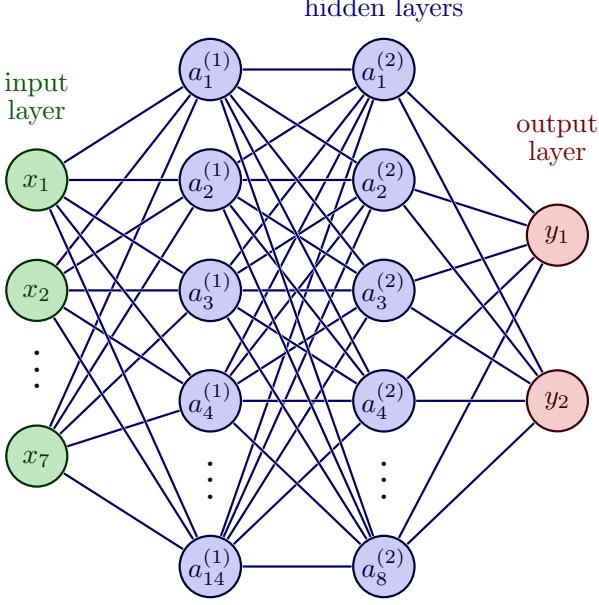

**Figure 7.** Sketch of the used classical fully connected ANN architecture. The architecture consisted of input nodes denoted as $x$, where $x_1 = \tilde{T}$, $x_2 = \tilde{M}^{1,7}(T))$, ..., and $x_7 = \tilde{M}^{6,12}(T))$. The hidden layers were represented by nodes denoted as $a$, and the output layer was denoted as $y$.

### 3.1.3. XANN: Towards Explainability

The architecture of the Explainable Artificial Neural Network (XANN) used in this study was taken from the work of J. Vaughan et al. [26]. The XANN architecture comprises an input layer, followed by a projection layer with a linear activation function. The projection layer is then connected to multiple subnetworks, where the number of subnetworks matches the number of nodes in the projection layer. Each node in the projection layer feeds its output to a separate subnetwork. All subnetworks have the same number of hidden layers and nodes and use the tangent hyperbolic function as the activation function (to cope with non-linear transfer functions, but any other activation function can be used). The outputs of the individual subnetworks are then merged in the output layer, which uses a linear activation function. The linear activation functions of the input, projection, and output layer are important to preserve the interpretability of the subnetwork transfer function, which should be the dominant function in the model. A sketch of such a XANN is displayed in Figure 8, while in Table 1, the number of weights and the activations for every layer are displayed. For a more detailed explanation of the theoretical background of the XANN architecture, please see [26].

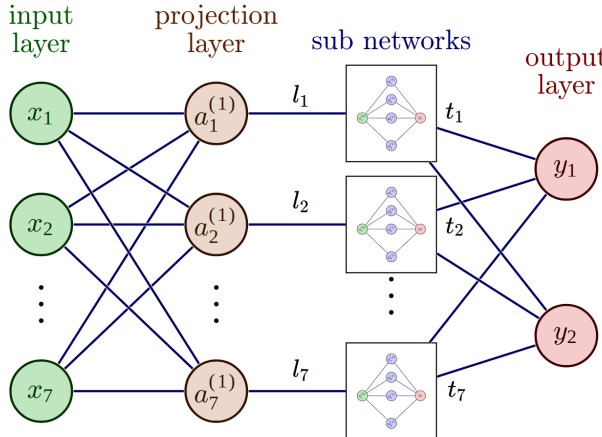

**Figure 8.** A sketch of the utilized Explainable Artificial Neural Network (XANN) architecture. The architecture consists of input nodes denoted as $x$, where $x_1 = \tilde{T}$, $x_2 = \tilde{M}^{1,7}(T))$, ..., and $x_7 = \tilde{M}^{6,12}(T))$. The input layer is fully connected to a projection layer with linear activation. The outputs of the projection layer nodes are then fed into separate subnetworks, where each projection node contributes its output $l_i$ to a specific subnetwork. Finally, the outputs of the subnetworks $t_i$ are merged in the nodes $y_1$ and $y_2$ of the output layer.

**Table 1.** Weight, activation, and shape of the XANN network layers.

| Layer | Number of Weights | Activation | Layer Shape |
|---|---|---|---|
| Input | 0 | Identity | Dense |
| Projection | 56 | Identity | one-to-one |
| Subnetworks' input | 0 | Identity | Dense |
| Subnetworks' hidden | 8 | Tanh | Dense |
| Subnetworks' output | 5 | Tanh | Dense |
| Output | 16 | Identity | - |

A XANN can provide information about the relations between the latent (projection layer output) $\vec{l}$ and model output variables $\vec{y}$, but not about the relations between the input $\vec{x}$ and the output $\vec{y}$ model variables, except if there is a nearly direct mapping of the input on the latent variables $\vec{l}$, i.e., each $x_i$ is straightly connected to the corresponding projection layer node $a_i$. But, the weights of the ongoing projection layer and outgoing subnetwork connections can provide information about the impact of different input variables and subnetworks, respectively, as well as the transfer functions $P_i(l_i) : l_i \rightarrow t_i$ of the subnetworks.

In this work, the XANN architecture comprises seven input nodes and a projection layer with seven nodes. The subnetworks are constructed with three layers, consisting of one, four, and one nodes, respectively. The activation function employed in the subnetworks is the tangent hyperbolic function. The output layer consists of two nodes, allowing the prediction of the X- and Y-position of the damage (or values close to zero indicating there is no damage).

All the neural networks (ANN and XANN) trained with UGW data universally employed the Adam optimizer with hyperparameters $\beta_1 = 0.9$ and $\beta_2 = 0.99$. Additionally, a fixed learning rate of $\alpha = 0.001$ was applied. These are the default hyperparameter values of TensorFlow; as they worked well, these default values were retained and not further optimized. The mean square error function (MSE) served as the chosen cost function for all neural networks.

### 3.2. Simple XANN Example

Although the XANN approach is promising to obtain at least a partial explainability of the predictor model, the following simplified example using the IRIS benchmark dataset

should demonstrate the possibilities of interpretation, limits, and issues with XANN models.

The IRIS benchmark dataset [28] consists of 151 sample instances, with four numerical input variables with values nearly in the same range and one categorical target variable (botanical classes *setosa*, *virginica*, and *versicolor*). Two of the three categorical classes can be separated by a linear discriminator function and the third only by a non-linear function, with respect to the input variables. The non-linearity of the model function can be clearly observed during the training, especially for the XANN, which is harder to be trained due to its subnetwork structure compared with a fully connected ANN. The training error reduces quickly, requiring only a few epochs covering the separation of the first two classes, but it also requires a high number of epochs to find subnetwork solutions that separate the third class. Additionally, a low update rate $\alpha = 0.02$ is required to find and converge to a suitable parameter solution. The training process using gradient error back-propagation is commonly highly unstable.

The playground architecture of the XANN is shown in Figure 9. After the network was trained (with an overall classification error below 4%, a typical value), the value range for all four latent variables (the output of the projection layer) was computed. The transfer function for each subnetwork is computed using the previously computed latent variable ranges plus an additional margin.

The results of the trained transfer functions of the subnetworks for three snapshot experiments are shown in Figure 10. All experiments used the same network architecture and data, but with the results of three independent trainings using a fixed-rate gradient descent error back-propagation method ($\alpha = 0.02$). Although the model parameter adjustment using error back-propagation is deterministic, the initial parameter values are initialized by a pseudo-random process delivering different start states. The results always show two linear and two non-linear subnetworks, but the specific assignment is arbitrary and mainly determined by the random parameter start state of the entire XANN. The polarity, the input and output value ranges, the offsets, and the slopes of the transfer functions can differ significantly and cannot be used for a general interpretation (explanation). This shows the curse of the high-dimensional (infinite) but sparse solution space. Each training ends up here in another local minimum in the loss-parameter space. As mentioned before, the training process is highly stepwise, as shown in Figure 11. This is a result of the independent and initially indistinguishable paths along the subnetworks. A small change in the parameters can create a big jump towards a suitable solution (high loss gradient).

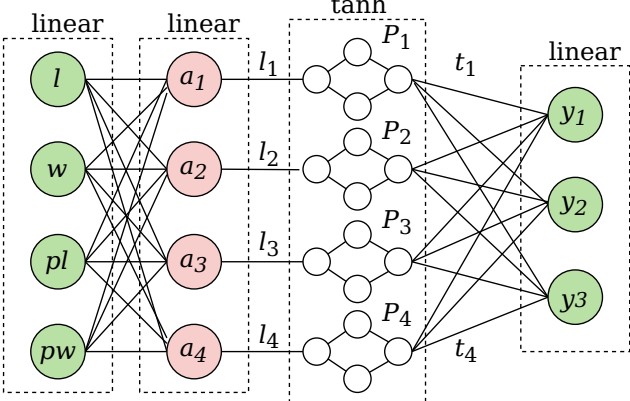

**Figure 9.** Playground XANN architecture for IRIS dataset classification. The input variables are *length*, *width*, *petal length*, and *petal width*. The output variables $\vec{y}$ are assigned to the three classes *versicolor*, *length*, and *virginica*.

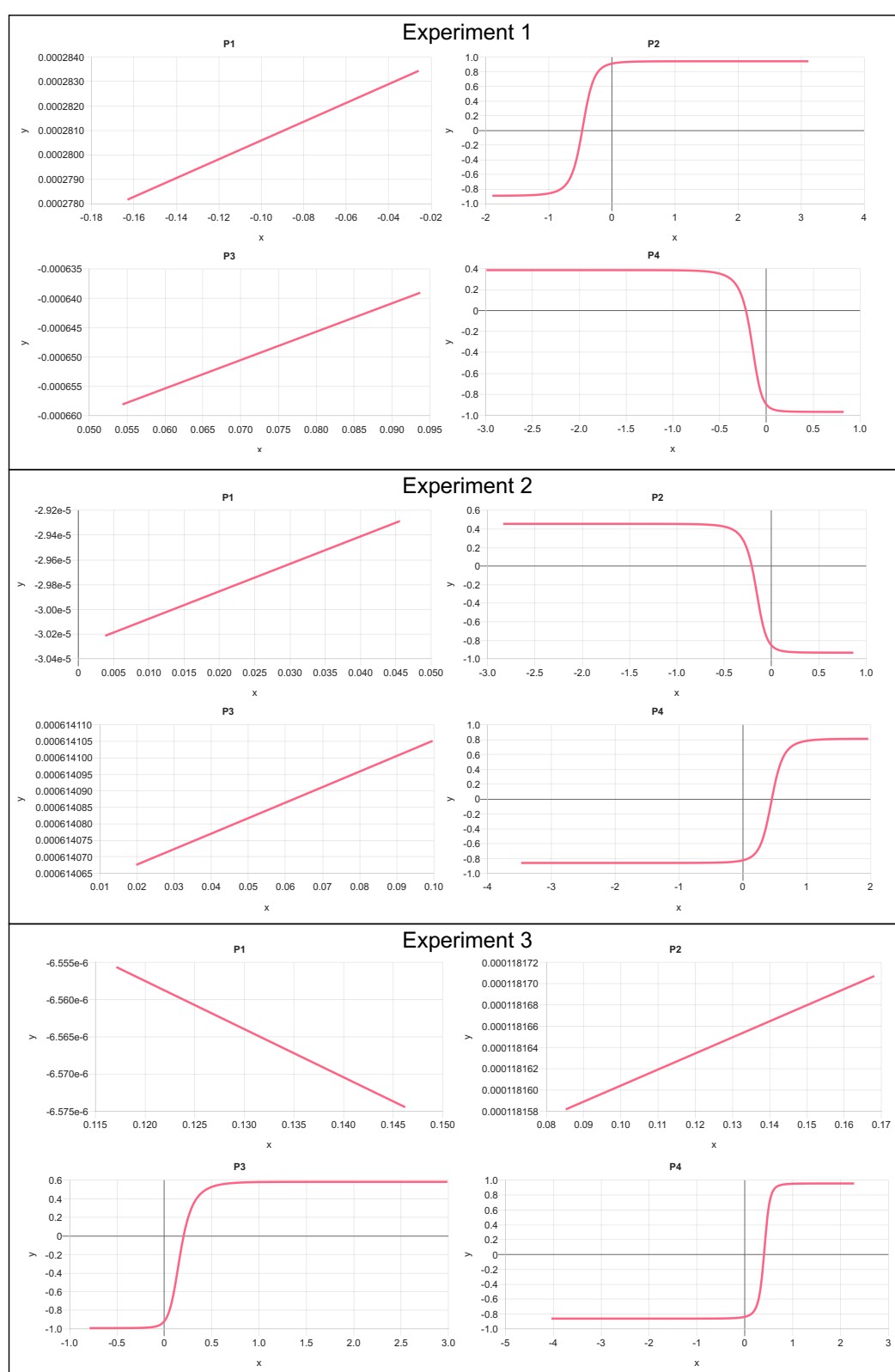

**Figure 10.** Subnetwork transfer functions from three independent experiments of the XANN for IRIS dataset classification (arb. units).

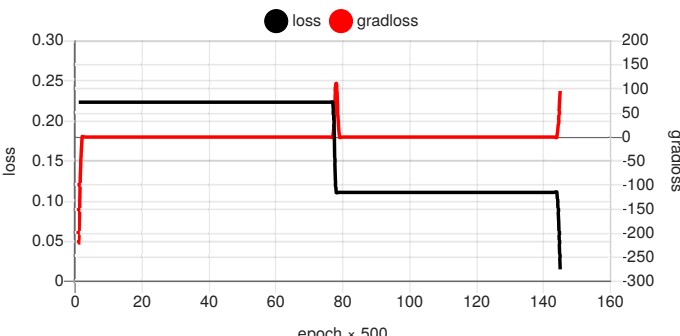

**Figure 11.** Example of the progress of the training with respect to the entire loss and the loss gradient (arb. units).

To obtain more insight into the classification performed by the XANN (separation of three different output classes), we have to consider the input weights of the output nodes ($y_1, y_2, y_3$, each assigned to one of the three classes). All three experiments obtain the same subnetwork selections:

- Output $y_1$ (class *versicolor*) uses both non-linear subnetworks with positive and negative gradients ($P_2, P_4$ for experiments 1, 2, and $P_3, P_4$ in experiment 3);
- Output $y_2$ (class *setosa*) always uses one subnetwork (negative gradient $P_2, P_4$, positive gradient $P_3$ in experiments 1, 2, and 3, respectively);
- Output $y_3$ (class *virginica*) always uses one subnetwork (positive gradient $P_2, P_4$, positive gradient $P_4$ in experiments 1, 2, and 3, respectively).

To summarize, we can conclude that mainly a superposition of non-linear subnetworks are used in all three experiments. The consideration of the input weights of the output nodes can be used to determine the impact of different subnetworks on the classification process. The polarity and the slope of the transfer function (i.e., the sign of the gradient) is irrelevant (because the sign and scaling value of weights can be arbitrary) and was used here only to distinguish them from each other. The linear subnetworks with their small output values and small slope can be neglected and are a hint for data variable reduction. A test can be made by setting the output to an average constant value and observing the classification results. If there is no degradation in the classification accuracy, these subnetworks can be removed (graph pruning).

*3.3. Noise and Prediction Errors*

Up until now, we have discussed the deployment of different ANN architectures to obtain insights into the functional mapping process behind a black-box predictor model using real or synthetic input data. Real measuring data is noisy, commonly given by a Gaussian statistical random distribution. Noise is either additive of multiplicative. Additive noise is generated by noise sources, e.g., by thermo-electrical effects and environmental disturbance. Multiplicative noise is commonly a result of measuring uncertainty. It is well known that data-driven models are noise-sensitive. Noise in training data can improve the generalization degree of the approximate model. Noise in prediction input data can increase prediction error or create false predictions. Often, false or incorrect predictions can be caused by a very low noise level. Well-known demonstrations exist in the domain of computer and automated vision [29], but any complex and highly non-linear models are affected by this noise sensitivity. Also, numerical noise (generated by computational functions) can result in incorrect predictions, as shown in [30] by Bosse et al. In their work, synthetically generated X-ray images were applied to a semantic pixel detector to mark material pores. Significant Gaussian noise was added to the X-ray training images. They observed that the predictor model produced artifacts (false positive markings) in different images at the same location (the model itself is totally location-independent), arguing that the low-level numerical noise of the X-ray image simulator (GPU-based, reduced numerical resolution) can be the cause of this phenomenon.

Robustness against noise is typically introduced during the training process by using augmented, noisy training data. If the measurement process is well known, the statistical noise generator can be set accurately. In this work, Gaussian noise is not added to the training data since we use computed features derived using an Analytical Signal via a Hilbert transformation, low-pass filters, and other non-linear transfer functions, creating complicated dependencies between the computed features and the original time-dependent sensor signals. That means the error propagation in the signal and time domains is unknown, as are the random processes of computed features.

The XANN model architecture can help to support the tracking of false model classifications and inaccurate regression predictions due to noise and its noise sensitivity by comparing the activation paths of training examples with the same noise-added variants. Misclassification due to low-level noise is typically an accumulative process along a set of highly non-linear functions (or function nodes in a graph).

But, in our work, simple CAR-ANN models are used. Actually, no low-level noise sensitivity can be observed, omitting the noise analysis by XANN. For any further more complex and non-linear models, the XANN noise analysis can be valuable.

## 4. Data Augmentation with Feature Functions

To apply the determined borders $\Delta M_{\text{up}}^{i,j}(T)$ and $\Delta M_{\text{low}}^{i,j}(T)$ for data augmentation, the first step involved generating a uniformly distributed temperature $T_{\text{s}}$ using a numerical random number generator following this rule:

$$20\,^{\circ}\text{C} \leq T_{\text{s}} \leq 60\,^{\circ}\text{C}. \tag{11}$$

The temperature $T_{\text{s}}$ was then used to generate an artificial maximum value $M_{\text{s}}(T_{\text{s}})$ using the following rule:

$$\Delta M_{\text{low}}(T_{\text{s}}) \leq M_{\text{s}}(T_{\text{s}}) \leq \Delta M_{\text{up}}(T_{\text{s}}). \tag{12}$$

Here, $M_{\text{s}}(T_{\text{s}})$ is also a uniformly distributed generated value using a numerical random number generator. In Figure 12, an example of the generated $M_{\text{s}}(T_{\text{s}})$ data is shown, using the borders from Figure 5. It can be observed that there are significantly more generated $M_{\text{s}}$ data points compared to the original $M$ data. With this method, depending on the numerical random number generator, any number of new data points can be generated. For instance, Figure 12 contains 100,000 generated data points, while the original measurements in Figure 5 consist of only 320 $M$ values. In general, for the subsequent experiments, 50,000 training data and 1000 test data were generated for each damage measurement, and 100,000 training data and 1000 test data were generated for the damage-free measurements for each transducer path considered.

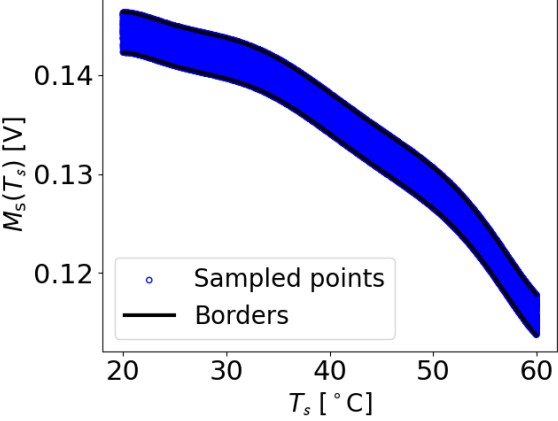

**Figure 12.** The same borders as in Figure 5, but with artificially sampled $M$ values generated using the established boundaries.

## 5. Results and Discussion

In this chapter, the network architectures and the different datasets are compared with each other, and the explainability of the XANN is examined. The relative error (RE) is calculated to compare the errors generated by the models. The absolute error $\sigma$ is calculated as the distance of the predicted damage position $[\tilde{x}, \tilde{y}]$ to the true damage position $[x, y]$:

$$\sigma = \sqrt{(\tilde{x} - x)^2 + (\tilde{y} - y)^2}. \tag{13}$$

RE is then calculated from $\sigma$ with respect to the largest error $\Sigma$ that can be measured on the test specimen. $\Sigma$ corresponds to the largest straight line that can be drawn on the test specimen. This is the diagonal distance from one of the lower corners to the opposite upper corner or vice versa and corresponds to approx. $\Sigma = 70.1$ cm:

$$\text{RE} = \frac{\sigma}{\Sigma}. \tag{14}$$

In Table 2, the mean RE over all damage cases of the different network architectures, trained and tested with both datasets, is shown. The models trained with the original data were trained for over 1000 epochs, not just 10 epochs, due to the limited amount of data. The hyperparameters were kept constant. It turns out that the errors of the models, in general, are not very large. This is probably due to the low variance in the damage positions (only four damage cases). It can be assumed that the errors of the models will increase with an increasing number of damage positions.

Another observation is that the models trained with the original data exhibit higher errors than the models trained with synthetic data. This applies to the original and the synthetic test sets. It can also be seen that the error of the models trained with synthetic data is approximately the same for both test sets, while the errors for the test sets of the models trained with the original data differ significantly. This suggests that the models trained with synthetic data are not only more accurate but also more stable than the models trained with the original data.

**Table 2.** Mean relative errors (MRE) over all damage cases for ANN and XANN architectures trained just with original OGW data or just with sampled artificial data.

| | ANN Original Training Data | ANN Sampled Training Data | XANN Original Training Data | XANN-Sampled Training Data |
|---|---|---|---|---|
| **MRE original test data** | 0.0062 | 0.00034 | 0.018 | 0.0015 |
| **MRE sampled test data** | 0.016 | 0.00034 | 0.03 | 0.0017 |

In the following sections, the different ANN architectures and different datasets will be investigated in more detail. For that, the errors of the models trained with synthetically generated data will be analyzed and compared in more detail. With that, it will be shown that models trained with these data have a low mean relative error (MRE). But, for the XANN, outliers occur with a higher deviation from the true damage positions (up to approx. 3% of $\Sigma$), which cannot be observed for the classic ANN. The XANN architecture will be used to examine these outliers and find the reason behind the high deviations.

### 5.1. Augmented Data for ANN Training

To show the capabilities of the artificially generated data for ANN model training for damage position prediction, the ANN were trained with the normalized and scaled input feature array $\tilde{I}$ with a batch size of 50 for 10 epochs, while the normalized and scaled original data, determined from the OGW data, served as the test dataset. In Figure 13, the results of the test predictions are shown for all damage cases. One can see that the ANN predictions are in close alignment with the true damage positions, so it is necessary to use the zoom (see Figure 13b–e) to observe the distribution of the predictions around the

damage position. It can also be seen that the model does not produce false negative or false positive results as well as the true damage case of the plate is always identified.

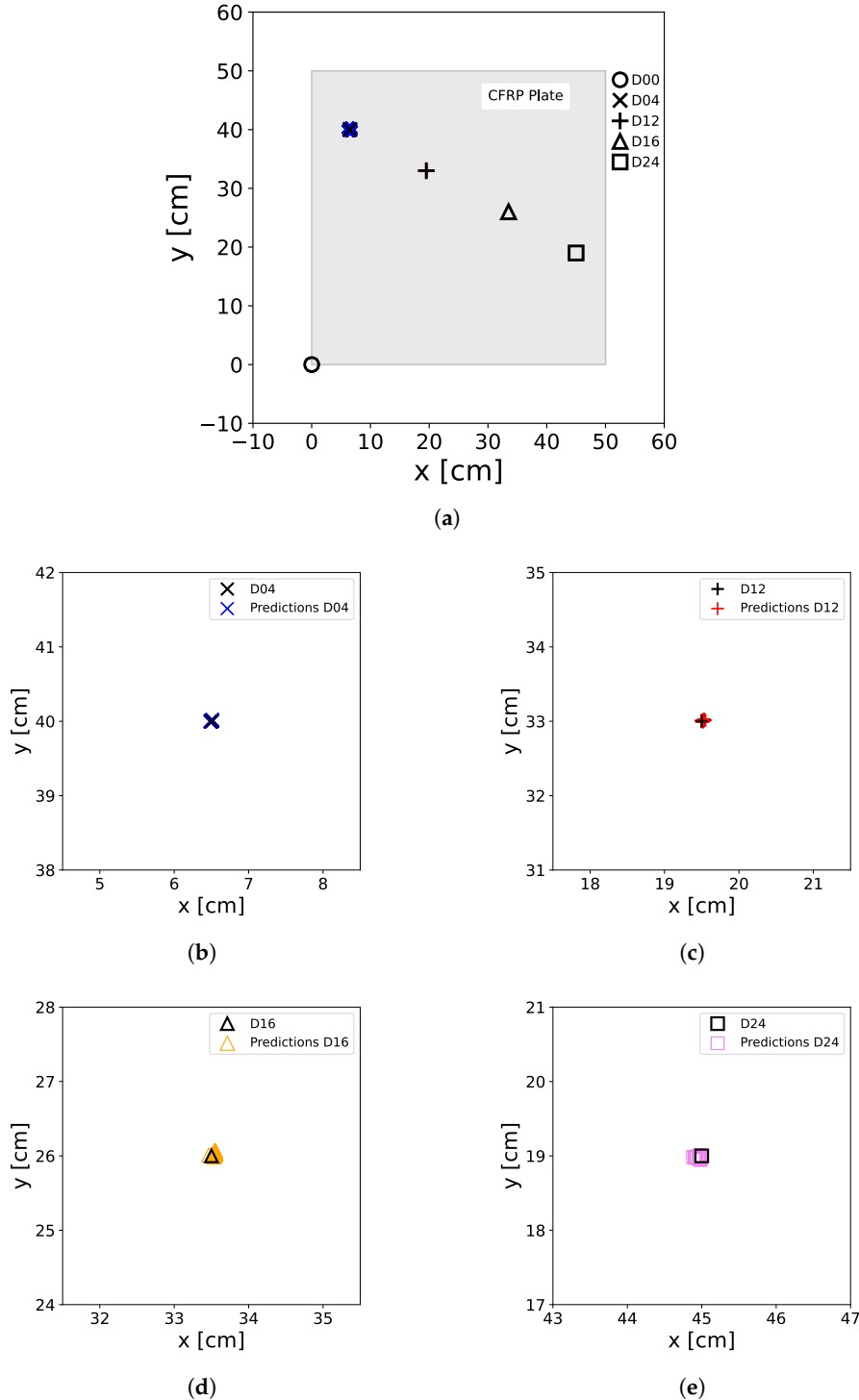

**Figure 13.** Predictions of the ANN trained with artificially generated data. For the predictions, the original *M* data extracted from the OGW dataset was used as a test set. The real positions of the damage on the plate, including D04 (×), D12 (+), D16 (△), and D24 (□), as well as the non-damage case D00 (○) at the [0,0] position, are represented by black markers. The colored markers illustrate the predictions made by the ANN, with D00 (○), D04 (×), D12 (+), D16 (△), and D24 (□) denoting the actual damage status of the input data used for prediction. (**a**) shows all damage predictions for the entire plate, including the damage-free case, while (**b**–**e**) show a zoom of the damage positions with the predicted positions.

In Table 3, the RE of the prediction results presented in Figure 13 is shown. It can be observed that the ER is very small, indicating high accuracy. This high accuracy is probably explained by the low variance and number of damage cases available for training.

**Table 3.** Relative errors for all damage cases for the classic ANN trained with artificially generated data. Errors were calculated from the test set predictions, which contained just original $M$ data.

| Damage Case | Mean RE | Maximum RE |
|---|---|---|
| D04 | 0.00016 | 0.00031 |
| D12 | 0.00026 | 0.00076 |
| D16 | 0.0005 | 0.0012 |
| D24 | 0.00043 | 0.0017 |

In the next step, the capability of the data augmentation method of reducing the number of necessary measurements while still producing enough training data is investigated by using an reduced set of UGW temperature measurements. Therefore, just data measured at temperatures $T = 20, 25, 30, 35, 40, 45, 50, 55, 60\,°C$ are used to compute the boundary functions $M_{up}^{i,j}(T)$ and $\Delta M_{low}^{i,j}(T)$, as shown in Figure 14. With these boundaries, artificially generated data can be seen in Figure 15.

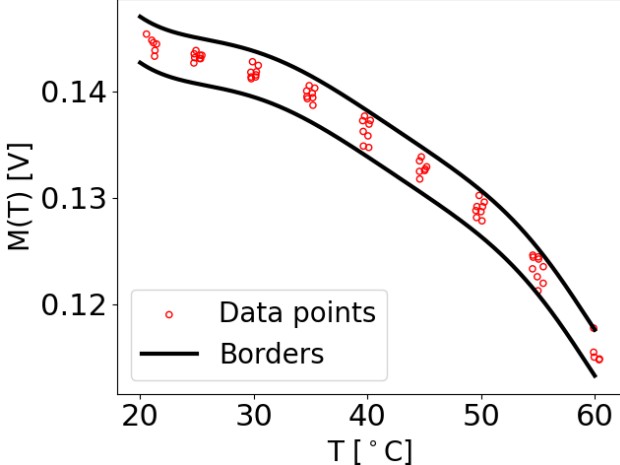

**Figure 14.** Examples of $M$ values extracted for $T = 20, 25, 30, 35, 40, 45, 50, 55, 60\,°C$, which were used to establish the borders from the maximum function $M^{i,j}(T)$, specifically for the damage-free measurement for the case where $i = 1$ and $j = 7$.

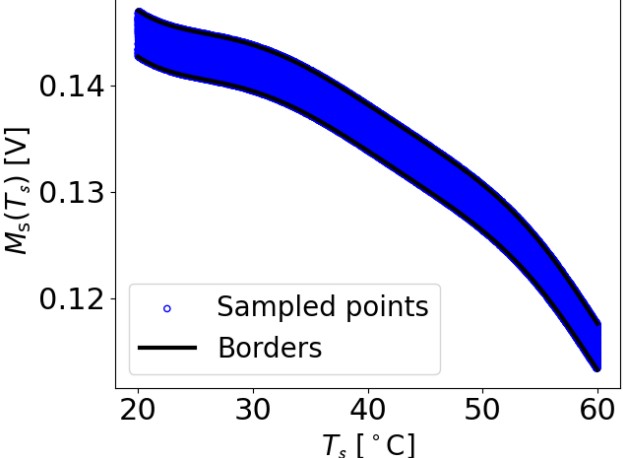

**Figure 15.** The same borders as in Figure 14, but with artificially sampled $M$ values generated using the established boundaries.

The dataset generated in this way was used as the training set for the ANN. The training was performed as described above. As the test set again, the original *M* data from the OGW dataset was used. In Figure 16, the predictions for the test set of the trained model are shown. Again, no false detections can be observed, and the predictions align close to the real damage positions. This shows that it is also possible to generate suitable artificial training data with the reduced UgW temperature dataset. But, it should be noted that one has to take care to use a step size between the temperatures that are not too large when using this approach. Since a large step size could result in data patterns that could be represented by other functions and therefore result in unsuitable artificial data.

Table 4 shows the ER of the predictions presented in Figure 16. One can see that the predictions of this model are still very accurate. However, upon comparison with Table 3, it becomes apparent that the errors in the model trained with artificial data and derived from the reduced set are marginally larger than those in the model trained with artificial data obtained from the complete measurement set.

**Table 4.** Relative Errors (RE) for all damage cases for the classic ANN trained with artificially generated data with original *M* data extracted at $T = 20, 25, 30, 35, 40, 45, 50, 55, 60\,^\circ$C. Errors were calculated from the test set predictions which contained just original *M* data.

| Damage Case | Mean RE | Maximum RE |
| --- | --- | --- |
| D04 | 0.00016 | 0.00052 |
| D12 | 0.00036 | 0.0016 |
| D16 | 0.00033 | 0.0019 |
| D24 | 0.0011 | 0.0011 |

With the classic ANN, our ability to examine the augmented data and assess its suitability for training ends. This is because we can only investigate what the ANN produces as output for a given input. In the following section, we aim to explore how this changes with the XANN architecture. This approach allows us not only to gain a better understanding of the prediction process within the trained XANN model, but also to examine how the model behaves concerning the given input data (here, original vs. augmented).

### 5.2. XANN

The XANN underwent training and testing following the same procedure as outlined for the ANN. The resulting predictions of the trained XANN can be seen in Figure 17. The performance is comparable to Figures 13 and 16. However, in the zoomed-in images of Figure 17, it can be observed that the predictions are more scattered around the damage locations compared to the normal ANN. It also shows greater errors. But, also with the XANN, no false predictions could be observed.

Table 5 displays the ER corresponding to the predictions shown in Figure 17. Remarkably, the errors in the XANN have increased compared to the classic ANN. However, they still fall within an acceptable range, reaching up to 0.5 % of $\Sigma$. It is noteworthy that the maximum ER deviates more from the mean ER in the XANN models than in the classical ANN models.

**Table 5.** Relative Errors (RE) for all damage cases for the XANN trained with artificially generated data. Errors were calculated from the test set predictions which contained just original *M* data.

| Damage Case | Mean RE | Maximum RE |
| --- | --- | --- |
| D04 | 0.00098 | 0.0044 |
| D12 | 0.00084 | 0.0027 |
| D16 | 0.0018 | 0.0027 |
| D24 | 0.0022 | 0.0052 |

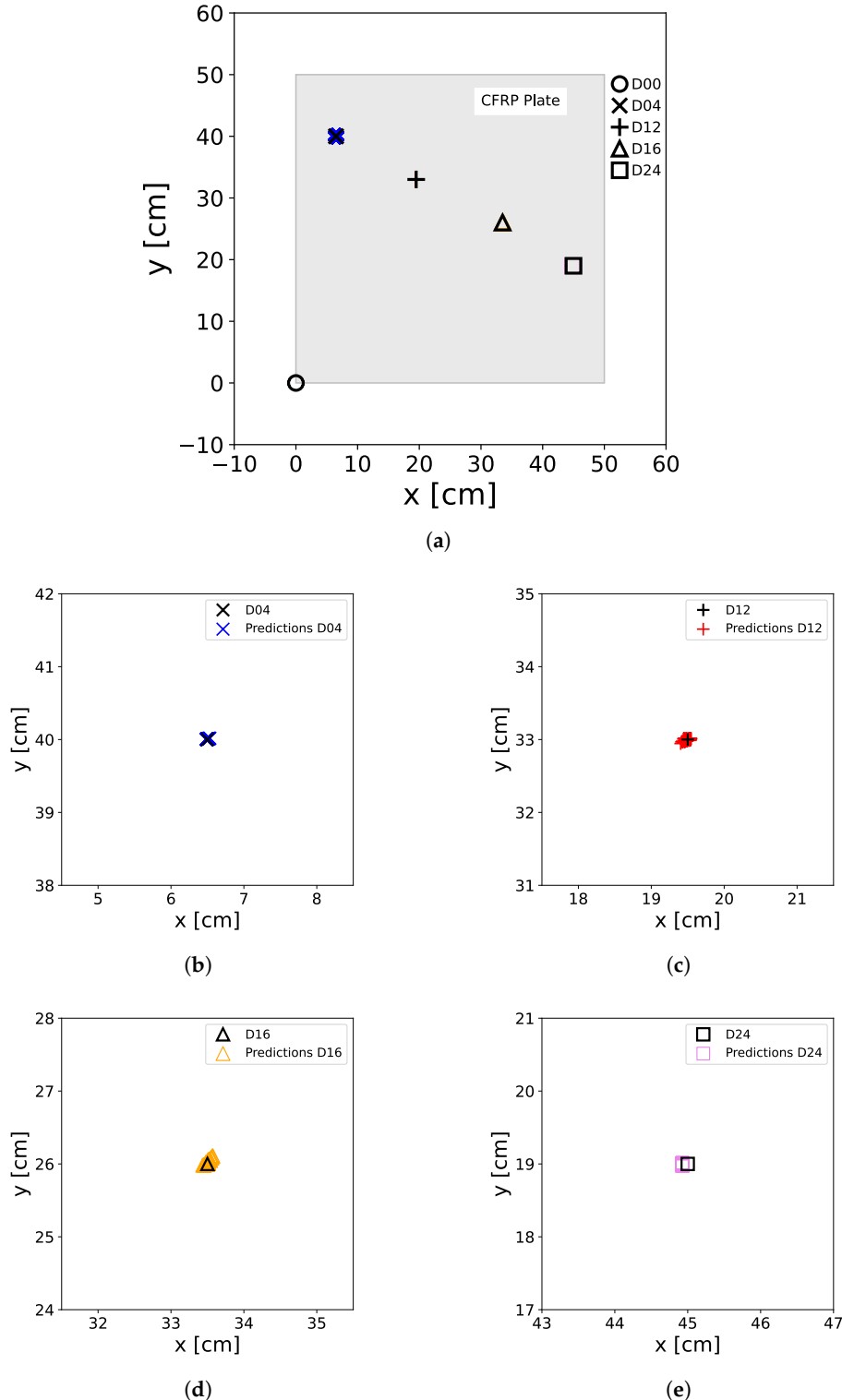

**Figure 16.** Predictions of the ANN trained with artificially generated data generated with original $M$ data extracted at $T = 20, 25, 30, 35, 40, 45, 50, 55, 60\,°C$. For the predictions, the original $M$ data extracted from the OGW dataset was used as a test set. The real positions of the damage on the plate, including D04 ($\times$), D12 ($+$), D16 ($\triangle$), and D24 ($\square$), as well as the non-damage case D00 ($\circ$) at the [0,0] position, are represented by black markers. The colored markers illustrate the predictions made by the ANN, with D00 ($\circ$), D04 ($\times$), D12 ($+$), D16 ($\triangle$), and D24 ($\square$) denoting the actual damage status of the input data used for prediction. (**a**) shows all damage predictions for the entire plate, including the damage-free case, while (**b**–**e**) show a zoom of the damage positions with the predicted positions.

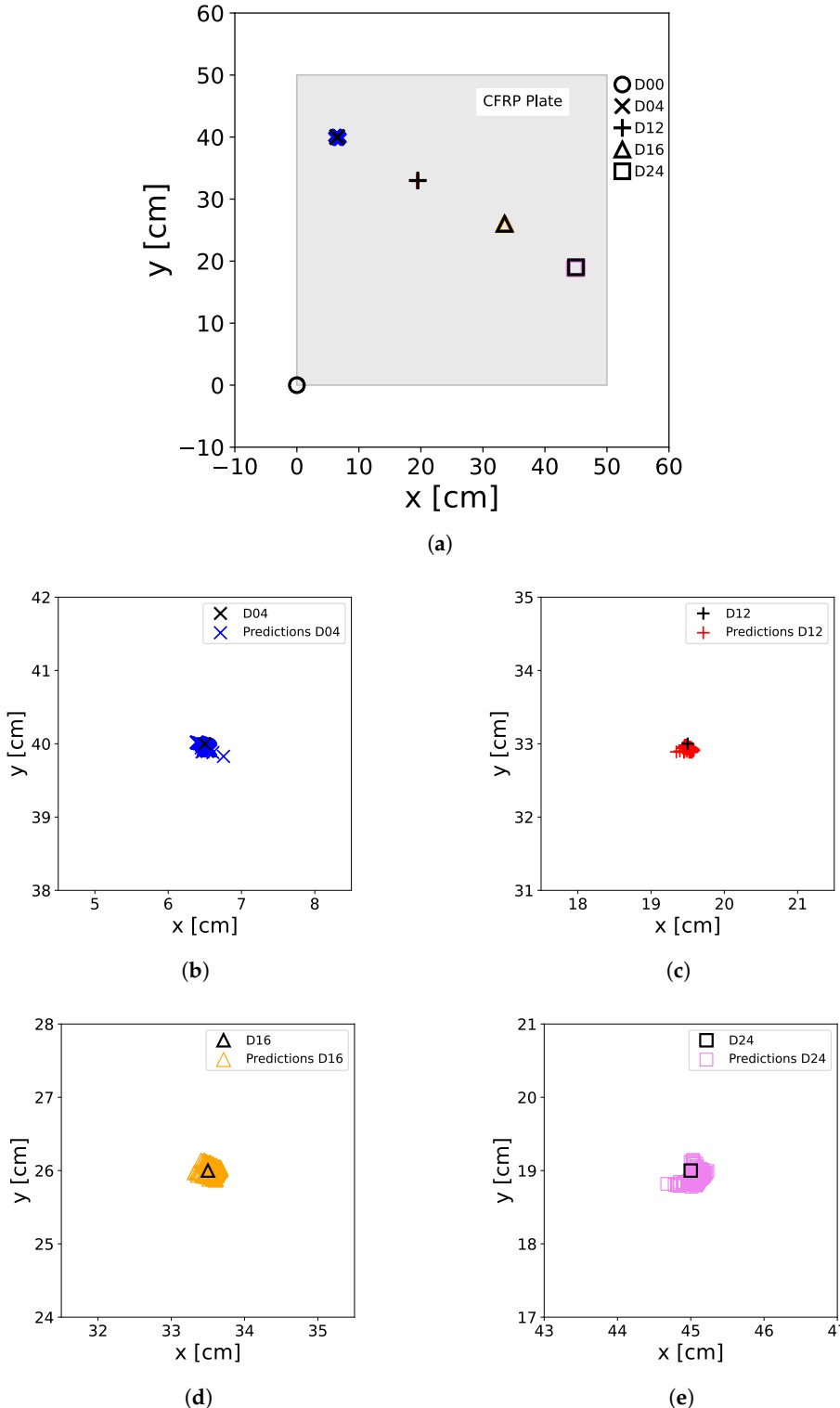

**Figure 17.** Predictions of the XANN trained with original data. For the predictions, the original *M* data extracted from the OGW dataset was used as a test set. The real positions of the damage on the plate, including D04 (×), D12 (+), D16 (△), and D24 (□), as well as the non-damage case D00 (○) at the [0,0] position, are represented by black markers. The colored markers illustrate the predictions made by the ANN, with D00 (○), D04 (×), D12 (+), D16 (△), and D24 (□) denoting the actual damage status of the input data used for prediction. (**a**) shows all damage predictions of the entire plate, including the damage-free case, while (**b–e**) show a zoom of the damage positions with the predicted positions.

However, to further explore the explanatory capabilities of the XANN, a test set was selected that included 1000 artificially generated feature values. This approach was adopted due to the fact that the initial test set does not encompass the entire spectrum of feature values present in the training set. A comparison between Figures 5 and 12 further illustrates this point. By using this test set, it can be demonstrated that the XANN is still able to accurately predict the damage status. The predictions of the XANN for this test set are depicted in Figure 18. However, in the zoomed-in images of Figure 18b–e, it can be observed that the distribution of the predictions around the damage positions is more scattered and has a larger error than the predictions in Figures 13, 16 and 17. The error can even be greater than 1 cm, resulting in outliers that are quite far from the point cloud of distributed predictions. Figure 18d highlights two of these outliers, which will be further examined later. Nonetheless, no false predictions were observed.

Table 6 presents the RE generated by the XANN model when utilizing the sampled data as input. As illustrated in Figure 18, the errors in this scenario show an increase, attributed to the higher variance present in the input data (compare, e.g., Figure 5 with Figure 12). The maximum error of D16 corresponds to outlier 1 from Figure 18d and is with approx. 3 % of $\Sigma$, which is much higher than the mean ER of D16. Even outlier 2, with an error of approx. 1.2 %, is relatively small compared to this.

**Table 6.** Relative Errors (RE) for all damage cases for the XANN trained with artificially generated data (same trained model as in Table 5). Errors were calculated from the test set predictions which contained just sampled $M$ data.

| Damage Case | Mean RE | Maximum RE |
|:---:|:---:|:---:|
| D04 | 0.0011 | 0.014 |
| D12 | 0.0011 | 0.015 |
| D16 | 0.002 | 0.027 |
| D24 | 0.0025 | 0.017 |

To investigate the XANN model's explanatory possibilities, the outputs of the subnetworks are studied as functions of their inputs. Since the input of each subnetwork is the output of the corresponding projection node, the input–output relationship can be visualized. For a reasonable visualization of the subtransfer functions, the minimum and maximum values of their inputs were computed. To generate input arrays for each subnetwork that cover the entire relevant value range of the transfer functions, the minimum and maximum were used as boundaries. The generated input arrays consist of 1000 data points and start with the minimum value and increase with every entry until the maximum is reached in the last entry of the array.

In Figure 19, the outputs of the subnetworks are plotted as functions of the input arrays, which demonstrates the explanatory capabilities of the XANN. The deeper analysis of the input–output relationship shows an interesting pattern.

One can see that the transfer functions of subnetworks 1, 2, and 7 seem to correspond to a quadratic function (subnetwork 1) or higher-degree polynomials (subnetworks 2 and 3). However, what is more remarkable and interesting is that subnetworks 3, 4, 5, and 6 correspond to shifted tangent hyperbolic and tangent hyperbolic functions multiplied by $-1$, respectively. This is not surprising since, as mentioned above, the networks use tangent hyperbolic functions as activation functions, whereas the shifts and the multiplication by $-1$ is probably explained by the learned weights of the nodes in the subnetworks. Later, we will see that the transfer functions of subnetworks 3, 4, 5, and 6 lead to strict separations of the output values with respect to the damage cases.

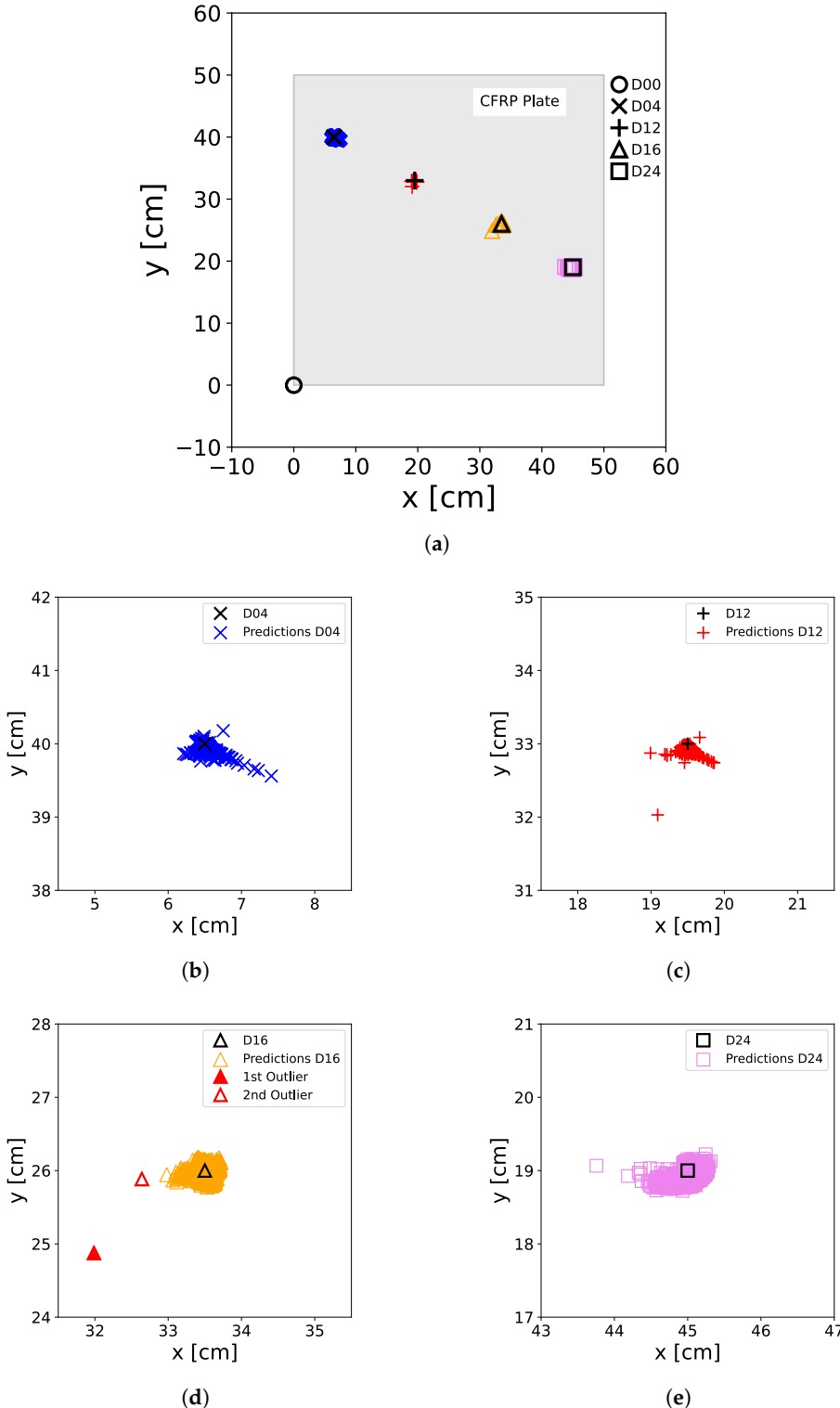

**Figure 18.** Predictions of the XANN trained with artificially generated data. For the predictions, a test set of artificially generated *M* data was used. The real positions of the damage on the plate, including D04 (×), D12 (+), D16 (△), and D24 (□), as well as the non-damage case D00 (○) at the [0,0] position, are represented by black markers. The colored markers illustrate the predictions made by the ANN, with D00 (○), D04 (×), D12 (+), D16 (△), and D24 (□) denoting the actual damage status of the input data used for prediction. (**a**) shows all damage predictions of the entire plate, including the damage-free case, while (**b**–**e**) show a zoom of the damage positions with the predicted positions. In (**d**), two outlier predictions are also marked.

Given that all output values fall within the range of −1 to 1, it is reasonable to expect that by appropriately weighting the output layer and summing the subnetwork outputs, values between zero and one can be generated. This range corresponds to the scaled positions of the damage labels (ranging from 0.2 to 1), while the damage-free case is represented by the [0,0] position. Thus, the XANN demonstrates its ability to predict the X and Y positions of the damage based on these generated values.

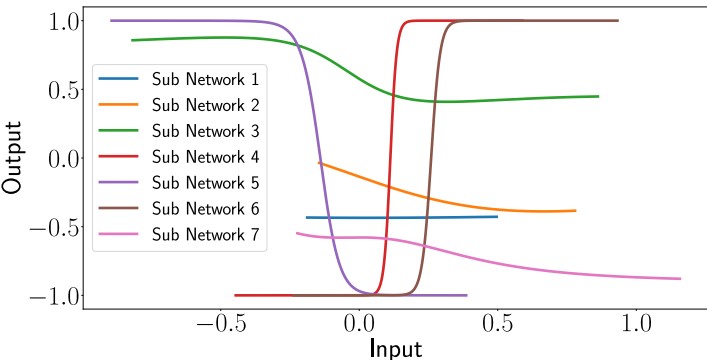

**Figure 19.** Outputs of the subnetworks of the XANN plotted as a function of their input values.

To gain more insights into the decision-making process of the XANN model, the outputs of the subnetworks are investigated with respect to the different damage cases. At first, the test set used in Figure 5 containing the original data was used as input for the trained subnetworks. Whereby the input features can be related to the corresponding damage cases. The images of these input–output plots are shown in Figure 18. Note that the range of input and output values differs from subplot to subplot.

To gain a deeper understanding of the processes in the XANN, the weights of the individual projection layer nodes with respect to the node inputs are listed in Figure 21. Since the individual projection nodes serve as input for the respective subnetworks, this approach allows us to examine the influence of the individual input features on the subnetworks. The subfigures in Figure 21 are arranged in such a way that, e.g., Figure 21a correspond to the weights of the nodes corresponding to the subnetwork from Figure 20a and so on.

For further investigation, subnetwork 5 will be presented and analyzed as an example in the following. A compressed examination of all the subnetworks can be found as bullet points in Appendix A.

In Figure 20e, the outputs of subnetwork 5 are displayed. One can see that the transfer function of subnetwork 5 corresponds to a shifted and negatively multiplied tangent hyperbolic function (violet line in Figure 20). Taking a closer look at Figure 20e, one can see that only the linear parts of the tangent hyperbolic function are used, which leads to a clear separation of the different damage cases. To be more precise, a separation of Damage D04 from the other damage cases is realized.

Turning our attention to Figure 21e, one can see that transducer path 6 → 12 is the most influential feature for subnetwork 5. Looking at Figure 2, it can be seen that damage D04 is located very close to transducer path 6 → 12, which results in a significant signal and feature modification in this transducer path due to this damage. Due to this spatial correlation of D04 and path 6 → 12, subnetwork 5 separates the damage D04 where the signal from transducer path 6 → 12 experiences a change (detects damage) from the cases where no or just small changes are measured in this transducer path (no damage is detected).

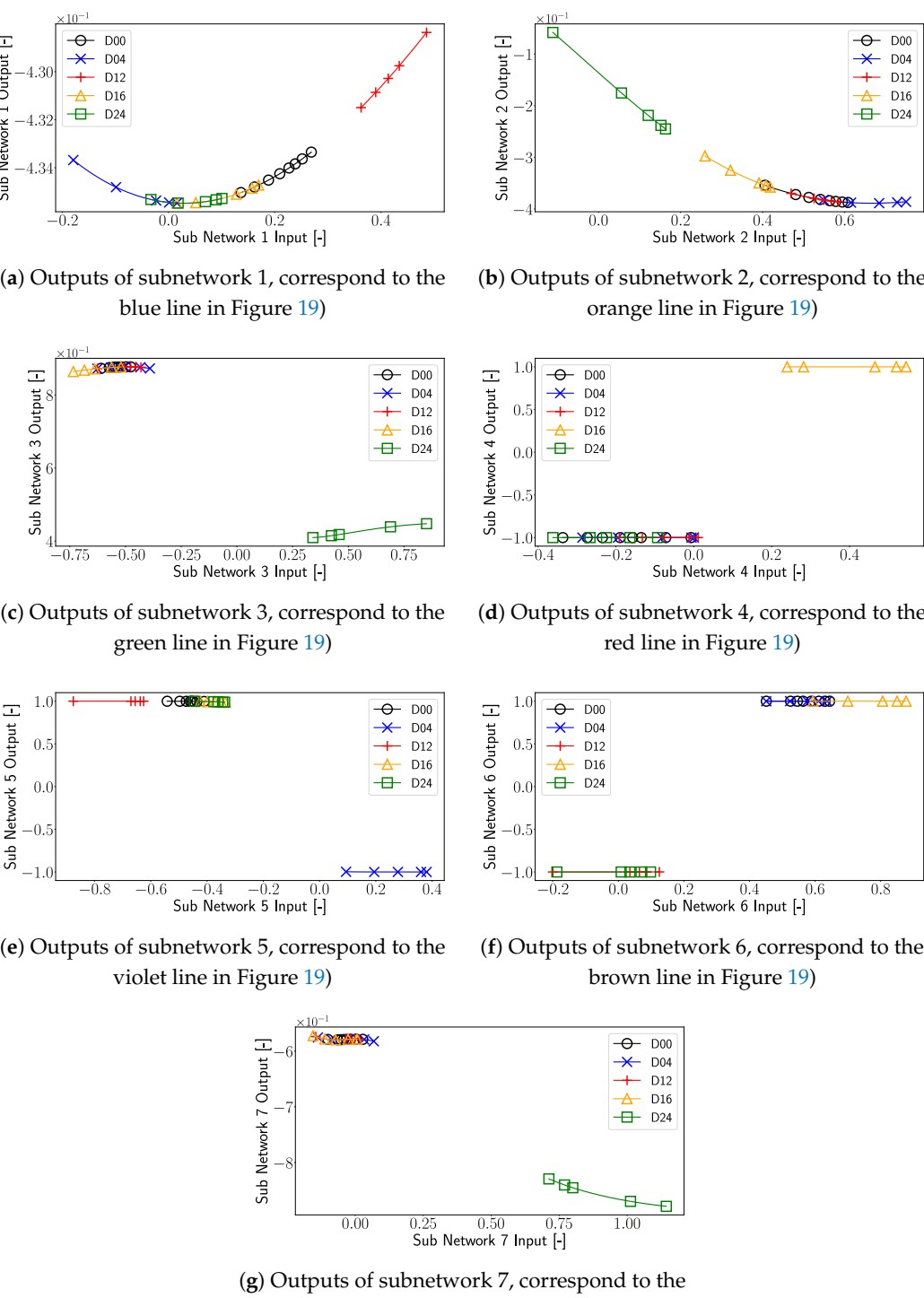

(**a**) Outputs of subnetwork 1, correspond to the blue line in Figure 19)

(**b**) Outputs of subnetwork 2, correspond to the orange line in Figure 19)

(**c**) Outputs of subnetwork 3, correspond to the green line in Figure 19)

(**d**) Outputs of subnetwork 4, correspond to the red line in Figure 19)

(**e**) Outputs of subnetwork 5, correspond to the violet line in Figure 19)

(**f**) Outputs of subnetwork 6, correspond to the brown line in Figure 19)

(**g**) Outputs of subnetwork 7, correspond to the pink line in Figure 19)

**Figure 20.** Outputs of the sub networks as a function of the original data as input. The markers used in the plot correspond to different damage cases of the plate, with D00 representing the damage-free case. In the interest of visibility for every damage case dataset, a marker was placed at the minimum and maximum data points, as well as at every 200th data point within these boundaries.

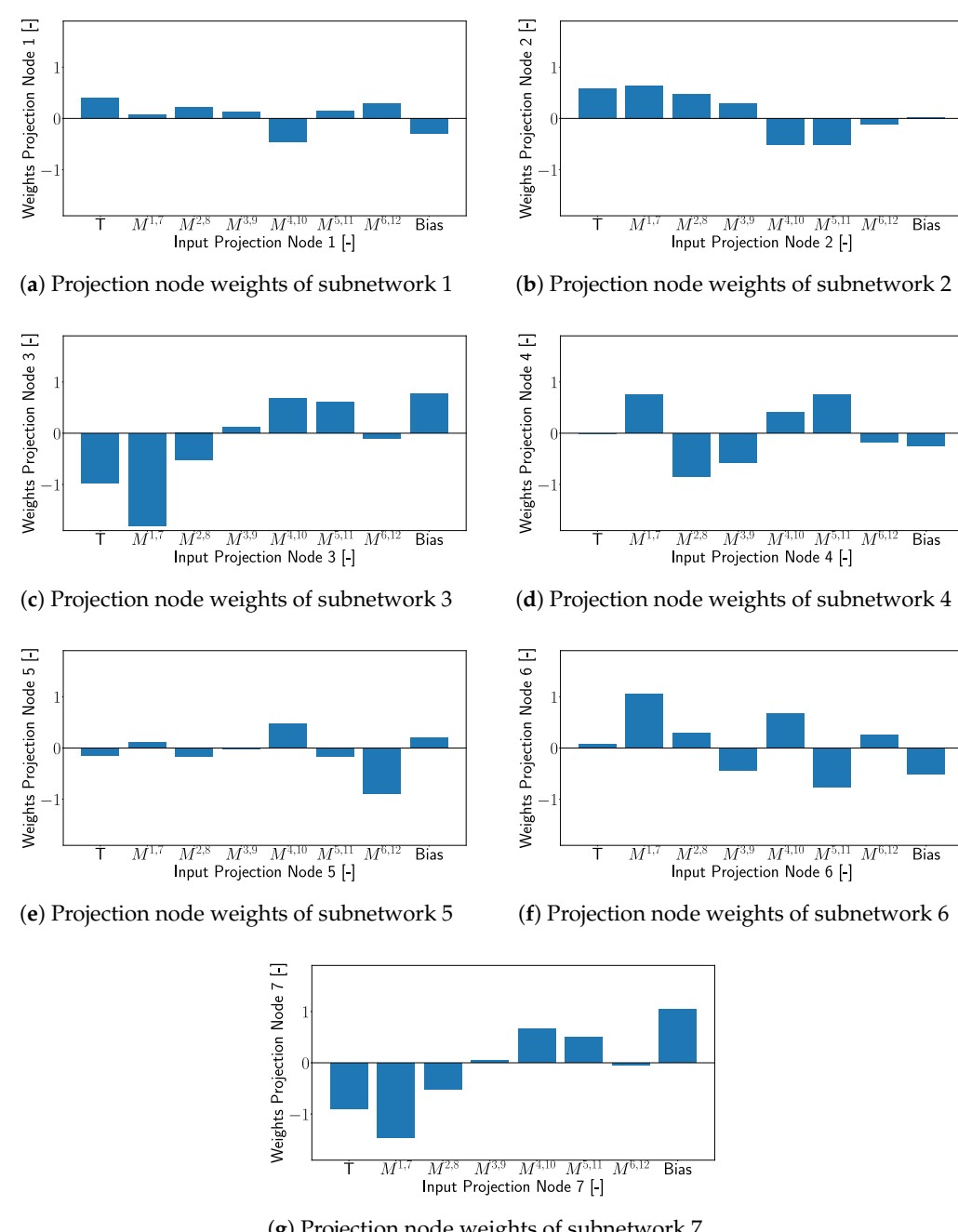

(**a**) Projection node weights of subnetwork 1

(**b**) Projection node weights of subnetwork 2

(**c**) Projection node weights of subnetwork 3

(**d**) Projection node weights of subnetwork 4

(**e**) Projection node weights of subnetwork 5

(**f**) Projection node weights of subnetwork 6

(**g**) Projection node weights of subnetwork 7

**Figure 21.** The wights of the projection layer nodes with respect to the inputs (corresponds to the XANN inputs). Where the notation $M^{i,j}$ corresponds to the $T$ dependent maxima of transducer path $i \to j$.

This indicates that for transducer path $6 \to 12$ that influences subnetwork 5 the most, the subnetwork differentiates between damage cases that can be detected from path $6 \to 12$ and those that cannot be detected from path $6 \to 12$. This pattern can be observed in all subnetworks with transfer functions similar to a varied tangent hyperbolic function, where each damage case could be clearly separated in at least one subnetwork from the cases where no damage was detected, as can be seen in Figure 20c–f. The separation of different damage cases, especially from the undamaged case, relies on the projection layer weights. These weights are important to identify the input features that have the most influence on the subnetwork and thus affects the influence of the transducer paths. Due to this damage, cases that are close to the most influential transducer paths of a subnetwork are more likely to be effectively separated from this subnetwork, as described in more detail in Appendix A.

In cases where the transfer function differs from a varied tangent hyperbolic function, the separation of the damage cases is less distinct and is more like a gradual transition. Similar to the linear transfer functions in Section 3.2, these transfer functions have only a small impact on XANN's prediction process.

It should be mentioned here that the shape and forms of the transfer functions of the subnetworks are different for every new trained model, as mentioned before in Section 3.2. This is due to the fact that for every training, different random initial weights are used, which leads to different influences of the input values on the subnetworks and, therefore, different transfer functions. But, it was shown that for every trained XANN model, some subnetworks applied a varied tangent hyperbolic function as a transfer function, similar to the results in Section 3.2. In Figure 22, the subnetwork transfer functions of a newly trained UGW data XANN are shown. As before, these transfer functions are sensitive to the most influential transducer paths and separate the damage cases where the damage was detected from these paths from those that were not detected from these paths. The influence of the inputs on a certain subnetwork is defined by the weights of the projection layer.

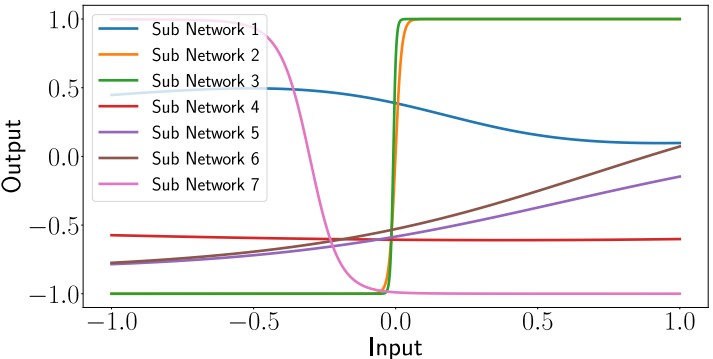

**Figure 22.** The outputs of the subnetworks in a newly trained second XANN model, trained with different initial random weights compared to the previous model, are plotted against their corresponding input values.

By analyzing the overall architecture of the XANN, one can see that the outputs of the subnetworks are used as inputs for the subsequent output layer and that the subnetworks constrain the inputs of the output layer within specific value ranges. It is important to remember that the labels of the x- and y-positions of the damage cases are scaled to values between 0 and 1, with [0,0] denoting the undamaged state and the activation function of the output layer corresponding to the identity function. Due to this, the output layer produces predictions of the damage positions by computing the linear combinations of the subnetwork outputs and their corresponding weights. In Figure 23, the weights of the output layer are shown, including the bias weights, which are also part of the linear combinations.

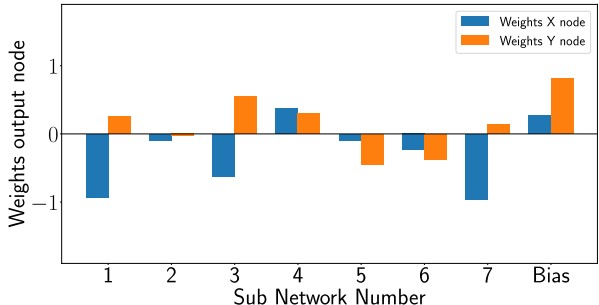

**Figure 23.** Weights of the output nodes, where the x node is responsible for predicting the x-position of the damage, and the y node is responsible for predicting the y-position of the damage.

In the following example, an approach is executed to gain a more profound understanding of how the output layer refines its predictions $P$. For that, in Table 7, the weight matrix $W$ of the output layer, and in Table 8, the rounded example outputs of the subnetworks' output vector $X$ for the undamaged case D00, have the bias node set to 1.

**Table 7.** Weights $W$ of the output layer.

| Subnetwork Number | $W$ x Node | $W$ y Node |
| --- | --- | --- |
| 1 | $-0.936$ | 0.253 |
| 2 | $-0.100$ | $-0.030$ |
| 3 | $-0.628$ | 0.551 |
| 4 | 0.381 | 0.305 |
| 5 | $-0.099$ | $-0.449$ |
| 6 | $-0.237$ | $-0.375$ |
| 7 | $-0.964$ | 0.135 |
| Bias | 0.264 | 0.821 |

**Table 8.** Example outputs of subnetworks for damage-free case D00.

| Subnetwork Number | Output $X$ for D00 |
| --- | --- |
| 1 | $-0.43$ |
| 2 | $-0.37$ |
| 3 | 0.88 |
| 4 | $-1$ |
| 5 | 1 |
| 6 | 1 |
| 7 | $-0.58$ |
| Bias | 1 |

Now the following operation is executed:

$$P = W^{\mathrm{T}} \cdot X, \tag{15}$$

where the symbol $\cdot$ represents the dot product and $W^{\mathrm{T}}$ represents the transposed weight matrix, while $X$ represents the vector of output values from the subnetworks. Using the output values from Table 8 and the weights from Table 7, we obtain the following values for the x and y positions:

$$x \approx -0.07 \, \mathrm{cm}$$
$$y \approx 0.0089 \, \mathrm{cm}.$$

These values are close to zero, and since no rescaling is performed for the undamaged case D00, this result was expected.

When one takes a closer look at the damage case D04, one can see that, in comparison with the undamaged case D00, only subnetwork 5 (see Figure 20e) shows a significant change in its outputs for D04. In this subnetwork, the output alters from 1 to $-1$, while for the other subnetwork, only minimal changes can be observed, so these changes are ignored in the following. This means that by changing the value of subnetwork 5 in Table 8 from 1 to $-1$, one should change the prediction of the output layer from D00 to D04. By repeating the operation from Equation (15) with the new subnetwork 5 value, the following values are obtained:

$$x \approx 0.19$$
$$y \approx 0.9.$$

Following the reverse scaling process, we attain

$$x \approx 6.1 \, \text{cm}$$
$$y \approx 40 \, \text{cm}.$$

Looking at Figure 2, one can see that these values are close to the true damage position of D04. This and the previous example show how the subnetwork outputs and the output layer weights achieve damage position predictions via linear operations.

Similar behavior can be observed in other damage cases. For example, for D12, only for subnetwork 5, the output value has to be changed from 1 to $-1$; for damage D16, just the output of subnetwork 4 has to be changed to modify the prediction from D00 to the corresponding damage case. For damage case D24, on the other hand, the output of three subnetworks has to be changed, specifically in subnetworks 3, 6, and 7, to switch from a D00 to D24 prediction. Subnetworks 1 and 2 remain unchanged for all damage cases, indicating that these subnetworks just have limited influence on the prediction-making process.

In Figure 23, it is evident that subnetwork 2 makes almost no contribution to the linear combination in the output layer since the corresponding weights are almost zero. Figures 20 and 20a, on the other hand, show that the outputs of subnetwork 1 are for all inputs almost constant, independent of the damage case so that no relevant change in the prediction can result from subnetwork 1.

These observations show that it is possible to gain deep insights into the decision-making process of a trained model with the XANN architecture.

In the next step, we want to investigate how one can use the XANN model to further validate if the artificially generated signal features are valid. For that, the outputs of the subnetworks of the trained XANN model are investigated. In Figure 24, the subnetwork outputs for the artificially generated signal features with respect to the damage cases are shown. By comparing Figure 24 with Figure 20, one can see that both Figures show in general the same pattern, with the difference that in Figure 24, the range of the inputs, and consequently the output values, are extended. This was expected since the artificially generated data itself has a larger value range than the measured signal features. For instance, in Figure 24d, this leads to the observation that subnetwork 4 is not only using the linear but also the non-linear part of the transfer function. For D16, the change in the minimum input value from approximately 0.2 in the original dataset to a minimum input value of about 0.14 in the sampled data caused this change. For the damage-free case, the shift is due to the change in the maximum input value from approximately 0 to around 0.1. In Figure 18a, one can see that this has no effect on the prediction for the damage-free case; however, this could be due to the lack of rescaling for this case. For damage D16, outliers can also be observed in Figure 18d, highlighted as first and second outliers.

In the next step, it will be investigated if the use of the non-linear components of the transfer function in subnetwork 4 is the reason for these outliers and if any peculiarities in the artificially generated data can be identified that can explain the outliers. In Figure 25, the normalized and scaled input values of each feature are plotted together with the mean input value along with its standard deviation. It can be seen that the input values for the outliers are significantly distant from the mean value in some cases (e.g., for the first outlier at T, $M^{1,7}$, $M^{2,8}$, and $M^{5,11}$) and that these deviations from the mean are larger than the standard deviations. But, it should be pointed out that other input values that do not result in an outlier show similar deviations from the mean value. Also, for other features, the input values of the outliers are quite close to the mean value. These observations indicate that the input features of the outliers do hold any unique position, i.e., they never represent the maximum or minimum input values or stand out in any particular way. Even if one analyzes the input data of the first outlier, which is the largest, no extraordinary position among the input values of transducer paths $2 \rightarrow 8$ and $2 \rightarrow 8$, which are the closest to D16, can be observed.

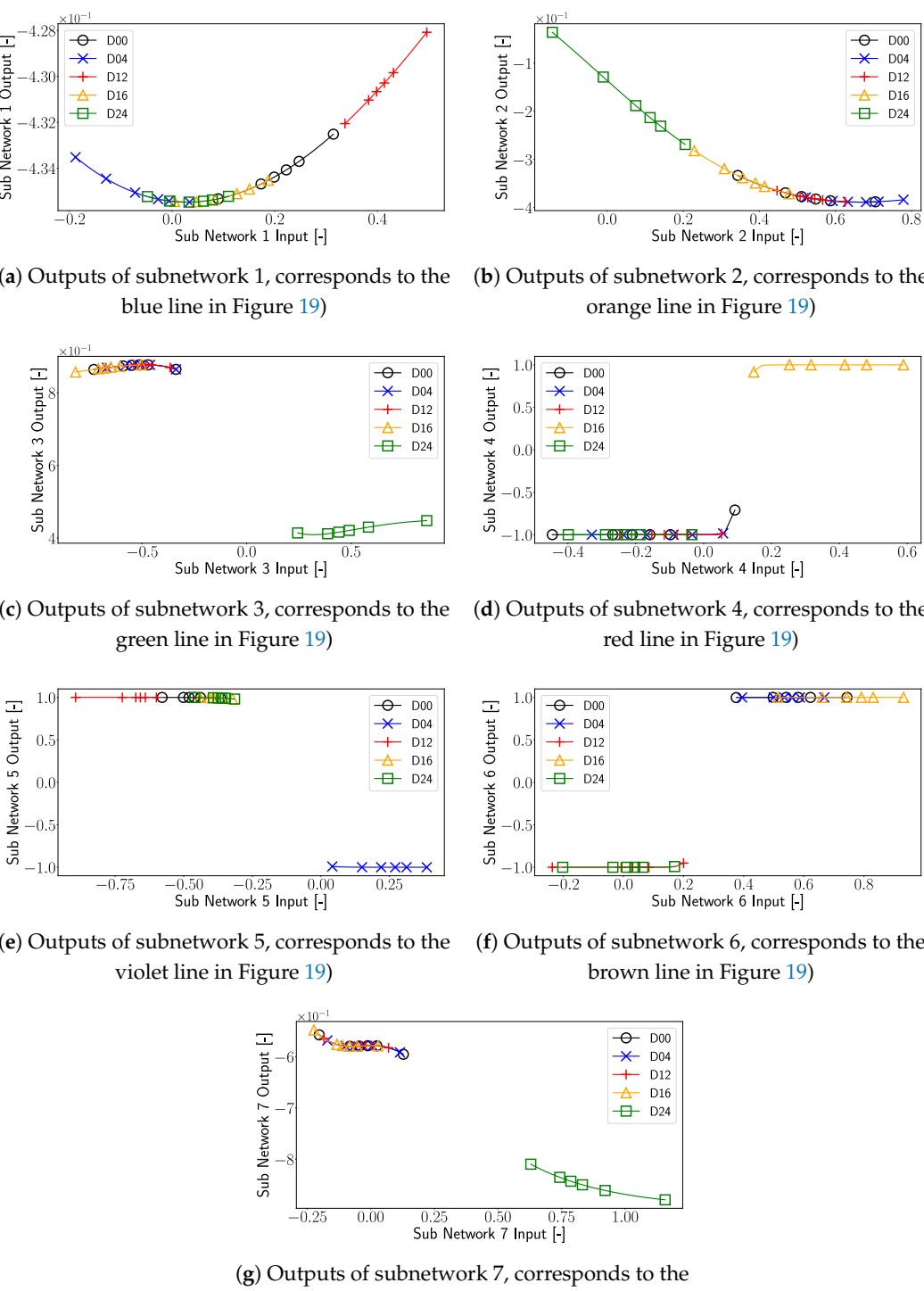

(**a**) Outputs of subnetwork 1, corresponds to the blue line in Figure 19)

(**b**) Outputs of subnetwork 2, corresponds to the orange line in Figure 19)

(**c**) Outputs of subnetwork 3, corresponds to the green line in Figure 19)

(**d**) Outputs of subnetwork 4, corresponds to the red line in Figure 19)

(**e**) Outputs of subnetwork 5, corresponds to the violet line in Figure 19)

(**f**) Outputs of subnetwork 6, corresponds to the brown line in Figure 19)

(**g**) Outputs of subnetwork 7, corresponds to the pink line in Figure 19)

**Figure 24.** Outputs of the sub networks as a function of the synthetic generated data as input. The markers used in the plot correspond to different damage cases of the plate, with D00 representing the damage-free case. In the interest of visibility for every damage case dataset, a marker was placed at the minimum and maximum data points, as well as at every 200th data point within these boundaries.

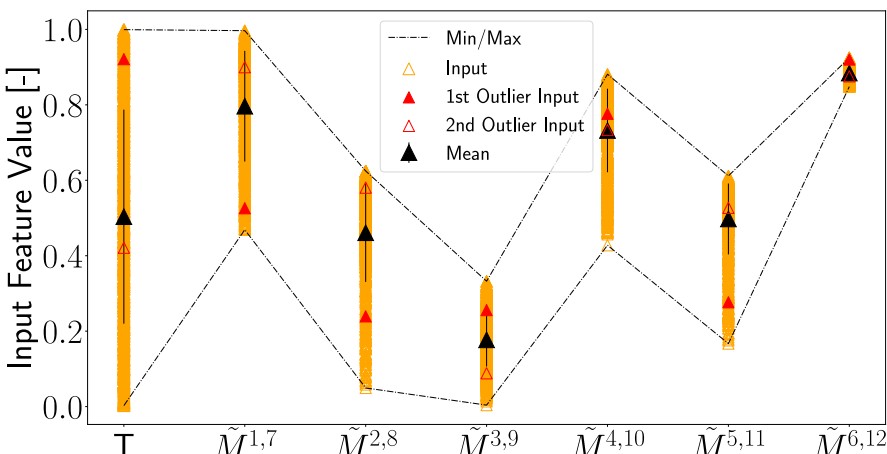

**Figure 25.** Feature input values artificially generated for damage case D16 are plotted. Shown are the mean value together with the standard deviation, as well as the minimum and maximum values of the input. The inputs for the first and second outliers, highlighted in Figure 18d, are plotted in a similar manner as in Figure 18d.

In Figure 26, which shows the inputs of the subnetworks (corresponding to the inputs in Figures 20 and 24), a different picture emerges, so that the input value of the first outlier for subnetwork 4 is the minimum input for subnetwork 4. Due to this, the output value for D16, which is located at the non-linear part of the transfer function (see Figure 24d), corresponds to the first outlier. But, also for the second outlier, unique positions can be observed since it represents the minimum input value in subnetworks 3 and 7. Particularly interesting is subnetwork 7, where the minimum input value of D16 leads to a significant deviation in the output. Looking at Figure 23, one can see that subnetwork 7 significantly influences the prediction of the x-position, which could explain why the second outlier deviates so much from the total damage position in the x-direction. The first outlier, on the other hand, shows large deviations in the x- and y-direction due to a similar strong influence of subnetwork 4 in the prediction of both directions; see Figure 23.

The inputs of the outliers hold no particular characteristics, as shown in Figure 25. But, on the other hand, the inputs of the outliers lead to significant changes in the subnetwork outputs. This implies that the composition of the outlier inputs represents a special case. Due to the fact that the individual feature values are generated independently, this is comprehensible since it could lead to input configurations that may not occur in reality so that the outputs of the subnetworks were shifted, resulting in a deviating position prediction.

Since the outliers are still quite close to the actual damage positions, it can still be assumed that the artificially generated features are valid. This also means that the assumption that a feature $M^{i,j}$ measured at $T$ lies between the boundaries $\Delta M^{1,7}\text{up}(T)$ and $\Delta M^{1,7}\text{low}(T)$ for each individual transducer path is valid. However, it is necessary to be careful when implementing the artificial generation of $M^{i,j}$ values across different paths.

It should be mentioned that the analysis carried out here could also be used to determine at which point a trained model fails when an individual transducer transmits incorrect data, e.g., due to malfunction.

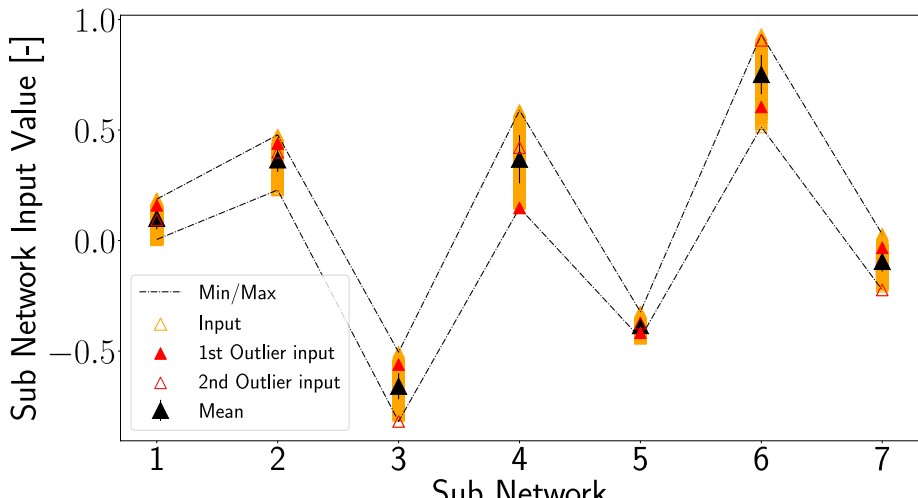

**Figure 26.** Inputs of the subnetworks produced by artificially generated input features for damage case D16 are plotted. Shown are the mean value together with the standard deviation, as well as the minimum and maximum values of the input. The inputs for the first and second outlier, highlighted in Figure 18d, are plotted in a similar manner as in Figure 18d.

## 6. Conclusions

In this work, a method for generating synthetic UGW feature data was developed and applied to an XANN framework. It was demonstrated that signal features, like the Hilbert envelope maximal values, can be represented as functions of variables like temperature. These functions allow it to generate new feature data using boundary functions, $\Delta M_{\text{up}}^{i,j}(T)$ and $\Delta M_{\text{low}}^{i,j}(T)$. These boundary functions define the range for creating new data points that lie between the boundaries. The success of this approach relies on the accurate determination of $\Delta M_{\text{up}}^{i,j}(T)$ and $\Delta M_{\text{low}}^{i,j}(T)$, which, in turn, depend on the availability of sufficient variance in the existing feature measurements. This approach should also work for other datasets if the specified conditions are met, although further validation via additional studies is essential.

Although the XANN architecture does not reveal the transfer function of the whole network, it provides valuable insights into the prediction process. This transparency helps to determine the impact of various input features on each subnetwork's output.

Our observations revealed that when damage occurs near sensor paths with substantial influence on a subnetwork, it results in the separation of that damage case within the subnetwork from other damage cases, particularly the undamaged case. It was also shown how the network predicted the damage positions of the damage cases with the help of the subnetwork outputs and the output layer weights.

Also, the validation of the synthetic-generated signal features was performed with the XANN by investigating outliers in the damage position predictions. It could be shown how the outliers appear in the trained XANN model. But, no special characteristics of the outlier input features could be found that could explain the outliers. Due to this, it is assumed that not a single "pathologic" input feature, but rather a specific composition of artificially generated input features, is responsible for the outliers. This indicates, in general, that the assumption that a feature lies between the boundary functions $\Delta M_{\text{up}}^{i,j}(T)$ and $\Delta M_{\text{low}}^{i,j}(T)$ is valid, but that the independent drawing can lead to problems.

These insights into the behavior of the XANN architecture hold the possibility of identifying when a specific trained model starts to falter. They also offer valuable guidance in the search for more efficient (X)ANN architectures and the identification of important signal features, since the XANN architecture has the capacity to determine which features truly influence the network's predictions. These research areas will be the focus of future studies.

**Author Contributions:** Conceptualization, C.P. and S.B.; methodology, C.P.; software, C.P.; validation, C.P.; formal analysis, C.P.; investigation, C.P.; resources, C.P. and A.S.H.; data curation, C.P.; writing—original draft preparation, C.P.; writing—review and editing, S.B. and A.S.H.; visualization, C.P.; general XANN analysis, S.B.; supervision, S.B.; project administration, A.S.H.; funding acquisition, A.S.H. and S.B. All authors have read and agreed to the published version of the manuscript.

**Funding:** This paper was funded within the Research Unit 3022 "Ultrasonic Monitoring of Fibre Metal Laminates Using Integrated Sensors" (Project number: 418311604) by the German Research Foundation (Deutsche Forschungsgemeinschaft (DFG)).

**Data Availability Statement:** The UGW data used in this article can be found at open guided waves [27] under https://doi.org/10.6084/m9.figshare.c.4488089.v1 (accessed on 2 February 2021). The programs used for data processing can be requested from the authors.

**Acknowledgments:** The authors expressly acknowledge the financial support of the research work on this article within the Research Unit 3022 "Ultrasonic Monitoring of Fibre Metal Laminates Using Integrated Sensors" (Project number: 418311604) by the German Research Foundation (Deutsche Forschungsgemeinschaft (DFG)).

**Conflicts of Interest:** The authors declare no conflicts of interest.

## Abbreviations

The following abbreviations are used in this manuscript:

| UGW | Ultrasonic Guided Waves |
|---|---|
| AE | Acoustic Emission |
| SHM | Structural health monitoring |
| ANN | Artificial neural network |
| XANN | Expandable neural network |
| OGW | Open guided waves platform |
| ToF | Time of Flight |
| OBS | Optimal Baseline Selection |
| BSS | Baseline Signal Stretch |
| CFRP | Carbon-fiber-reinforced plastic |

## Appendix A

*Appendix A.1. Examination Subnetwork 1*

- Subnetwork 1 demonstrates damage case separation using a quadratic function in both input and output (Figure 20a);
- The projection layer handles input value separation;
- The subnetwork manages output separation;
- Clear separation is observed for D04 and D12 from the undamaged case (D00). D16 even overlaps with the undamaged case;
- Figure 21a shows that no input significantly influences subnetwork 1;
- Inputs from transducer paths $4 \rightarrow 10$ and $6 \rightarrow 12$ have slightly more influence;
- Damage cases D04 and D12 are close to these paths in Figure 2, indicating their influence and explaining their separation in subnetwork 1.

*Appendix A.2. Examination Subnetwork 2*

- Subnetwork 2 uses a quadratic function for damage case separation (Figure 20b);
- D24 is clearly distinguished from D00, while other damage cases overlap with D00;
- In Figure 21b, most input weights have similar absolute values, indicating a similar level of influence on subnetwork 2, except for paths $3 \rightarrow 9$ and $6 \rightarrow 12$;
- The weight for path $1 \rightarrow 7$ is slightly higher;
- In Figure 2, D24 is located on this path, showing its significant influence on this path's signal, explaining D24's effective separability in subnetwork 2.

*Appendix A.3. Examination Subnetwork 3*

- Subnetwork 3's transfer function is a compressed, shifted, and negatively multiplied tangent hyperbolic function (Figure 19);
- This function results in a distinct and abrupt separation of damage cases (Figure 20c), separating damage cases that are indistinguishable from the undamaged case from those where damage can be detected;
- In contrast to subnetwork 2, where D04 and D16 partially overlap with D00, subnetwork 3 exhibits complete overlap of these damage cases with D00 (making the damage cases indistinguishable from the undamaged case);
- Figure 21c shows that transducer path $1 \rightarrow 7$ has a significantly notable influence on subnetwork 3;
- Combined with D24's substantial influence due to its positioning on this transducer path, the pronounced distinctiveness of D24's signal features in subnetwork 3 becomes evident.

*Appendix A.4. Examination Subnetwork 4*

- Subnetwork 4 employs a shifted tangent hyperbolic function as its transfer function, leading to a distinct and abrupt separation of damage cases (Figure 20d);
- There is an even more pronounced jump between damage cases compared to subnetwork 3 due to subnetwork 4's use of an unstretched tangent hyperbolic function;
- In subnetwork 4, D16 is distinctly separated from other damage cases, while D04, D12, and D24 overlap with D00;
- Figure 21d shows that no single input dominates others in subnetwork 4;
- Inputs from transducer paths $2 \rightarrow 8$ and $3 \rightarrow 9$ collectively exert a significant influence on the subnetwork;
- D16, positioned between these transducer paths, exhibits strong separability;
- D12 is located between paths $4 \rightarrow 10$ and $5 \rightarrow 11$ which, due to their weights, should also have significant influence, raising questions about the distinguishability of D16 over D12 in this context;
- It is known that, unlike D12, D16 also influences path $1 \rightarrow 7$. Taking this into account, D16 is expected to exert the most influence on the subnetwork, providing an explanation for its separation from the other damage cases.

*Appendix A.5. Examination Subnetwork 5*

- Subnetwork 5 distinctly separates D04 from other damage cases (Figure 20e);
- It uses a shifted and negatively multiplied tangent hyperbolic function for this clear separation;
- Figure 21e reveals that transducer path $6 \rightarrow 12$ has the most significant influence on subnetwork 5;
- D04 is positioned directly on this transducer path, where it generates a strong sensor response, making its separation by subnetwork 5 highly likely.

*Appendix A.6. Examination Subnetwork 6*

- Subnetwork 6 utilizes a shifted tangent hyperbolic function as its transfer function;
- Figure 20f reveals distinct separation of two damage cases, D24 and D12;
- Despite their spatial separation, Figure 21f shows that transducer paths $1 \rightarrow 7$, $4 \rightarrow 10$ and $5 \rightarrow 11$ have significant influence on subnetwork 6;
- D24's direct positioning on path $1 \rightarrow 7$ results in a robust signal response, explaining its separation in subnetwork 6;
- D12, located between paths $4 \rightarrow 10$ and $5 \rightarrow 11$, generates substantial signal responses on two pathways due to its positioning, contributing to its favorable separation in subnetwork 6.

*Appendix A.7. Examination Subnetwork 7*

- Subnetwork 7 utilizes a transfer function that appears to resemble a higher-degree polynomial;
- Figure 20g shows a clear separation of D24 from other damage cases;
- Figure 21g highlights that transducer path $1 \rightarrow 7$ has the most significant influence on subnetwork 7;
- This influence explains the prominent separation of D24 compared to other damage cases within subnetwork 7.

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
