# Peer review of "Damage Location Determination with Data Augmentation of Guided Ultrasonic Wave Features and Explainable Neural Network Approach for Integrated Sensor Systems"

_computers, doi:10.3390/computers13020032_

Round 1

Reviewer 1 Report

Comments and Suggestions for Authors

The structure of the article is incorrect, first of all, there is no separate chapter containing conclusions. The elements of the conclusions are mixed with the discussion of the results, it is difficult to separate and analyze them.

To make matters worse, it is not clear whether the authors' goal was to show the usefulness of XANN for interpreting a function approximated by a neural network or to propose a method that allows multiplication of the number of learning patterns.

If the goal was to propose a method to increase the number of learning patterns, then this should be the focus of the conclusions and the advantages of the method should be clearly presented. The example of applying XANN to a biology-related task should then be removed and the usefulness of XANN demonstrated in a different way.

If the goal was to show the usefulness of XANN then the introduction should be prepared differently and the title of the article should be changed, the purpose of using XANN should be clearly shown and discussed in the 'Final remarks' section.

The discussion presented in the article is not, in the reviewer's opinion, readable in terms of the resulting conclusions. Does the recognition of the internal structure of data processing by XANN allow us to draw any conclusions about the preprocessing of data? Or the selection of the most relevant inputs of the network? Or can any other advantages useful from the point of view of network application can be demonstrated?

In any case, the introduction (state of the art) must include more references to the literature, citing only 9 items (half of which are own publications) is clearly insufficient. Whether in the field of SHA or guided ultrasonic waves, the literature is extremely rich.

In the reviewer's opinion, the structure of the article should be rebuilt, the main objective should be chosen, the introduction should be matched to the main objective and the summary (conclusions) should be completed. Only then will it be possible to write a proper review.

Author Response

Comments and Suggestions for Authors

The structure of the article is incorrect, first of all, there is no separate chapter containing conclusions. The elements of the conclusions are mixed with the discussion of the results, it is difficult to separate and analyze them.

-> Conclusions were added, also the manuscript is re-structured

To make matters worse, it is not clear whether the authors' goal was to show the usefulness of XANN for interpreting a function approximated by a neural network or to propose a method that allows multiplication of the number of learning patterns.

-> XANN's serve multiple goals in this work: 1. An architecture that allows to investigate the relationship between input and output variables (only partially possible, but improved compared with ANN); 2. As a predictor model architecture, too; 3. To identify critical paths inside the network, e.g., proving noise robustness (discussed in the new Section "Noise and ...") ; 4. Improve and support the model-assisted synthetic data augmentation.

If the goal was to propose a method to increase the number of learning patterns, then this should be the focus of the conclusions and the advantages of the method should be clearly presented. The example of applying XANN to a biology-related task should then be removed and the usefulness of XANN demonstrated in a different way.

If the goal was to show the usefulness of XANN then the introduction should be prepared differently and the title of the article should be changed, the purpose of using XANN should be clearly shown and discussed in the 'Final remarks' section.

-> A major part of the manuscript is rewritten in order to make things more clear and to show how the artificial data sampling benefits from the used XANN model 

The discussion presented in the article is not, in the reviewer's opinion, readable in terms of the resulting conclusions. Does the recognition of the internal structure of data processing by XANN allow us to draw any conclusions about the preprocessing of data? Or the selection of the most relevant inputs of the network? Or can any other advantages useful from the point of view of network application can be demonstrated?

-> The discussion is rewritten and shows how the XANN can be used for input data inspection, in terms how the output of the subnetworks change when using artificial data instead of measured data. And with that giving a tool to investigate when the rained model starts to fail.  

In any case, the introduction (state of the art) must include more references to the literature, citing only 9 items (half of which are own publications) is clearly insufficient. Whether in the field of SHA or guided ultrasonic waves, the literature is extremely rich.

-> Introduction is rewritten and more references are included

In the reviewer's opinion, the structure of the article should be rebuilt, the main objective should be chosen, the introduction should be matched to the main objective and the summary (conclusions) should be completed. Only then will it be possible to write a proper review.

-> see points above

Reviewer 2 Report

Comments and Suggestions for Authors

Please find attached the pdf document with the comments. Thank you.

Comments on the Quality of English Language

Author Response

REVIEW
Damage Location Determination with Data Augmentation of Guided
Ultrasonic Wave Features and Explainable Neural Network Approach for
Integrated Sensor Systems
Authors: Christoph Polle, Stefan Bosse and Axel S. Herrmann

The paper presents a methodology to classify different types of damage by using different machine learning
techniques. Data augmentation is performed on a temperature-variable open-access dataset (Open Guided
Waves), analyzing the maximum values of the envelope of the signals, and creating artificial data with the
definition of boundaries for the analyzed data. Later, the expanded dataset is fed to an Artificial Neural
Network (ANN) with some sub-networks before the output layers which enable the explainability of the
phenomena, converting the network into an XANN (X stands for Explainable). This XANN is able to classify
both real and artificial data and, thanks to the subnetworks, to explain the output of the network and how it
processes each sample to decide to which class corresponds. Moreover, the paper is correctly organized.
Overall, the work is interesting to the NDT/SHM community, since it presents novelty in the applied
methodology. However, it presents several major issues that must be addressed before being considered for
publication.

Major issues:

* English must be carefully analyzed. Many sentences must be rewritten in order to make the text more
comprehensible, since it is sometimes difficult to read. Please enhance it.

-> A major part of the manuscript is rewritten and should be more comprehensible and better to understand

* The paper presents 4 sections (Introduction, Materials and Methods, Results and Discussion), whose
content is clear and well contextualized within the context. However, the lack of the Conclusion section
(which must be added), as well as some redundant content, makes the text difficult to follow. Please
reorganize the content, add some subsections (e.g., a proof-of-concept section for the IRIS benchmark,
transfer functions, and weights comparison subsections, etc.), and consider summarizing the content in
some sections (e.g., analysis of Figures 19 and 20 sounds redundant).

-> We have revised the paper structure to avoid repetition and added a conclusion section. In addition, more results were discussed to clarify the relationship between the two topics (data augmentation and XANN) and to support our conclusions with results.

* The introductory section is very short and does not present important papers in the field of damage
detection and localization using machine learning. A paper of this type cannot only have 9 references
(being two of them references to the dataset, and two self-citations). Please add and analyze more
literature (especially related to machine learning and SHM).

-> We have revised the introduction part and included more sources

* The first list in the introduction section is not universal. There are many other possibilities of applying
SHM. Please check Farrar’s book “SHM: A Machine Learning Perspective” for the SHM paradigms
and the objectives (detection, localization, quantification, classification, prognosis).

-> We adjusted the list as suggested to the paradigms in 'SHM: A Machine Learning Perspective'.

* It would have been very interesting to also analyze the frequency changes with the temperature. The
frequency content considerably shifts when the temperature varies, so the signal characteristics also
vary. The reviewer suggests considering this effect in future works, but it should be mentioned in the
text.

-> "In previous works, we have thoroughly investigated this (see: 'Polle, C.; Bosse, S.; Koerdt, M.; Maack, B.; Herrmann, A.S. Fast Temperature-Compensated Method for Damage Detection and Structural Health Monitoring with Guided Ultrasonic Waves and Embedded Systems. In Proceedings of the Advances in System-Integrated Intelligence; Valle, M.; Lehmhus, D.; Gianoglio, C.; Ragusa, E.; Seminara, L.; Bosse, S.; Ibrahim, A.; Thoben, K.D., Eds.; Springer International Publishing: Cham, 2023; pp. 362–378.'). In these works, we found that this dataset exhibits the highest sensitivity to damages at 40 kHz. Therefore, in the present work, we decided to focus on this frequency to keep the scope of the work manageable."

* Equations (5) and (6) suggest that deltaM_up and deltaM_low are constants, but the final graph boundaries are not
consistent with this effect. Please carefully check these values throughout the text (check in lines 445-
456).

-> In fact, DeltaM_up and DeltaM_low are constants, while DeltaM_up(T) and DeltaM_low(T) as defined in equations (5) and (6) are temperature-dependent limit functions, which are generated by adding or subtracting the DeltaM_up and DeltaM_low constants. There is probably some confusion here as the notation of the functions is very similar to that of the constants. To eliminate this confusion, we have now labeled the constants with a small delta while the functions are labeled with a large delta. In addition, the text describing how the functions DeltaM_up(T) and DeltaM_low(T) are calculated has been revised to create clarity between the functions and the constants.

* Why do the authors use hyperbolic tangent activation functions? And sigmoid in the output layers? This
must be justified in the text, as well as the hyperparameters
and values.

-> tanh was used as one of the common activation functions due to its non-linearity and its use for separation off different classifications (non-linear separation space). The sigmoid activation was used in the original ANN output layer because the labels were scaled between 0 and 1 which are the upper and lower boundaries of the sigmoid function. In the XANN the identity function was used in Input, projection and output layer since this layers should not be the dominant functions of the XANN. The hyper parameters used for the ANN and XANN employed the default values of tensorflow, since these parameters worked well, no hyper parameter optimization was done. The hyper parameter for the toy XANN were chosen by experience and were related to common ANN implementations targeting the IRIS dataset problem.

All these information are in the manuscript text now.  

* The authors should include more information on the proposed network: number of weights, shape of the
layers, activations, etc. As an example, TensorFlow shows a summary of the network with these features.
Please consider to include.

-> a table with these information was added

* Figure 10 and the associated text are difficult to understand. Please check the text. The reviewer also
suggests reducing the size of the graph.

* Figures 12, 15, etc. showing the damage localization results can be substantially enhanced, by zooming
not only damage D12 but also the others. Please rearrange these figures.

-> Figuers were rearranged and the zoom of all damage positions are shown now

* The discussion section reads only a qualitative analysis. Quantitative analysis should be added, in order
to reduce the redundant text and to have a numerical vision of the transfer functions and weights
influence.
-> We have reduced the redundancy by rearranging the chapters and through exemplary deeper analysis. We have also added the errors of the predictions to better analyze the accuracy of the network models. 

* Figure 6, vertical values are presented as dimensionless, but they should be squared volts.

-> Now A has dimension V²

Round 2

Reviewer 1 Report

Comments and Suggestions for Authors

The authors took into account all comments included in the first review. I have no further comments.

Author Response

Changes have been made based on the comments of the second reviewer.

Reviewer 2 Report

Comments and Suggestions for Authors

Please find attached a .pdf document with the reviewer's comments. Thank you

Comments on the Quality of English Language

Author Response

•English should be analyzed again. Several sentences are redundant. Moreover, some phrases appear
twice in the text (i.e., looks like a copy-paste, e.g., lines 263 and 379).

-> English is analysed and redundant prases are changed or erased

•The introductory section should explain a little bit more each added reference. In addition, some
important references can still be added. As an example, references for data-driven methods for
temperature compensation can include:
o Yue, N., & Aliabadi, M. H. (2020). A scalable data-driven approach to temperature baseline
reconstruction for guided wave structural health monitoring of anisotropic carbon-fibre-reinforced
polymer structures. Structural Health Monitoring, 19(5), 1487-1506.
o Azuara, G., & Barrera, E. (2020). Influence and compensation of temperature effects for damage
detection and localization in aerospace composites. Sensors, 20(15), 4153.

-> References are added and all important references are more expained now

•Add the SHM acronym to the abstract, as well as the uppercase letters. Consider changing GUW to
UGW (like in Rose’s book [9], more common nomenclature, might increase searches).

-> SHM acronym and uppercase letters are added to the abstract, GUW is changed to UGW.

•It would be clarifying to add some pictures from the original article of OGW showing the real
experimental set-up.

-> Picture from original article was added, also a short decription of the UGW measrement process was added.

•In figure 17(d), the outlier is considered as a prediction which is further than 1 cm from the actual
position. However, in figure 17(e), it seems that there is another outlier at x=44, y=19. Please clarify
this.

-> You are correct that there is another outlier in Figure 17(e); this is also the case for Figures 17(b) and (c). 
The outliers in Figure 17(d) were merely used as examples to demonstrate the capabilities of the XANN architecture, 
as we have two clearly visible and distinguishable outliers here. However, the choice of these outliers should not 
imply that there are no outliers in other damage scenarios.

Round 3

Reviewer 2 Report

Comments and Suggestions for Authors

The authors have addressed all my suggestions. The paper is now ready for publication.